# EVALUATING THE INSTRUCTION-FOLLOWING ABILITIES OF LANGUAGE MODELS USING KNOWLEDGE TASKS

## ABSTRACT

In this work, we focus our attention on developing a benchmark for instruction-following where it is easy to verify both task performance as well as instruction-following capabilities. We adapt existing knowledge benchmarks and augment them with instructions that are a) conditional on correctly answering the knowledge task or b) use the space of candidate options in multiple-choice knowledge-answering tasks. This allows us to study model characteristics, such as their change in performance on the knowledge tasks in the presence of answer-modifying instructions and distractor instructions. In contrast to existing benchmarks for instruction following, we not only measure instruction-following capabilities but also use LLM-free methods to study task performance. We study a series of openly available large language models of varying parameter sizes (1B-405B) and closed source models namely GPT-4o-mini, GPT-4o. We find that even large-scale instruction-tuned LLMs fail to follow simple instructions in zero-shot settings. We release our dataset, the benchmark, code, and results for future work.

## 1 INTRODUCTION

The growth of increasingly powerful large language models has resulted in the development of end-user applications including assistants for coding and software engineering (Ozkaya, 2023; Zhang et al., 2023; Ross et al., 2023), workflow and business automations (Grohs et al., 2023; Wornow et al., 2024), self-help assistants (Zhou et al., 2020; Shuster et al., 2022) and more. The need for highly accurate and controllable systems that follow precise instructions have led to the development of methods to improve reliability and consistency in the output for LLMs. Such methods include few-shot prompting (Gao et al., 2020; Kojima et al., 2022b), reasoning with explanations (Wei et al., 2022; Huang & Chang, 2022), checking for consistency/self-consistency (Wang et al., 2022), use of intermediate evaluators or LLMs operating as judges (Zheng et al., 2023), etc.

While there has been a lot of focus on assessing the knowledge of LLMs (Brown et al., 2020; Heinzerling & Inui, 2021), logical reasoning (Hendrycks et al., 2021; Wei et al., 2022; Ma et al., 2024), programmatic ability (Dakhel et al., 2023; Chen et al., 2021), problem solving ability (Lightman et al., 2024), etc, the study of their ability to follow precise instructions is relatively nascent; works such as FoFo (Xia et al., 2024), InFoBench (Qin et al., 2024), RuleBench (Sun et al., 2024), IFEval (Zhou et al., 2023b) attempt to address this gap. While FoFo Xia et al. (2024) assesses the ability of models to generate outputs conforming to existing real-world output formats such as the HL7-CDA format used in Healthcare applications, RuleBench (Sun et al., 2024) assesses a model's capabilities on inferential rule-following using rules which can be encoded in instructions and first-order-logic. On the other hand, benchmarks such as InFoBench (Qin et al., 2024) and IFEval (Zhou et al., 2023b) assess the ability of LLMs to follow arbitrary task specific instructions though neither InFoBench nor IFEval provide easy ways of verifying (i) task success and (ii) instruction following capabilities simultaneously (see Section 2 for a detailed discussion). Constraints such as formatting style, length are harder to verify along with task performance (not just instruction performance) while assessing instruction-following with verfiable is limited.

In this work, we focus our attention on developing a benchmark for instruction-following where it is easy to verify both task performance as well as instruction following capabilities. We adapt existing

Table 1: Comparison with existing Instruction Following benchmarks

| Benchmark | Task | Deterministic Outputs | Content Verification | QA-Conditioned Instructions | Evaluator |
|---|---|:---:|:---:|:---:|:---:|
| FoFo Eval (Xia et al., 2024) | Format following | ✓ | ✗ | ✗ | LLM |
| IFEval (Zhou et al., 2023b) | Instruction Following | ✓ | ✗ | ✗ | Direct |
| InFoBench (Qin et al., 2024) | Instruction Following | ✗ | ✓ | ✗ | LLM |
| RuleBench (Qin et al., 2024) | Inferential Rule Following | ✓ | ✓ | ✓ | Direct |
| This work | Instruction Following | ✓ | ✓ | ✓ | Direct |

commonly used knowledge benchmarks including MMLUPro (Wang et al., 2024), MathQA Amini et al. (2019), Winogrande (Sakaguchi et al., 2021), BoolQ (Clark et al., 2019), PIQA Bisk et al. (2020) and augment them with two broad classes of instructions: (i) Instructions that are conditional on the answer to the question (ii) Instructions that are applied uniformly regardless of the answer or task. We include a detailed study of multiple LLMs and find that even the largest models have trouble following relatively simple instructions. Our list of instructions demonstrate: 1. Simple changes of return text instead of labels results in a drop. 2. Simple tasks of counting, concatenation, conditional exclusion/inclusion/application as well as distracting instructions all result in significant drop in performance. To the best of our knowledge, there is no prior work that demonstrates this with verifiable results and bench-marking of current models.

Our contributions are as follows: (i) We release the first benchmark that assesses the zero-shot instruction-following performance of models using knowledge and reasoning question-answering (QA) tasks. (ii) We employ multiple QA-conditioned instructions to examine instruction-following performance across different instruction classes, including those dependent on answer-type. (iii) We include instruction instances that serve as distractors for the original knowledge-tasks (iv) Unlike previous studies, we use LLM-free evaluation metrics to assess both knowledge and instruction-following abilities. (v) We offer automated error analysis measures, pre-classifying likely errors for each instruction instance. (vi) Our benchmark creation method is easy to extend to new instructions and datasets.

## 2 RELATED WORK

Evaluating the capabilities of large language models (LLMs) has been a significant area of research, with studies focusing on various aspects of LLM performance. Researchers have developed multiple benchmarks to assess factual knowledge Petroni et al. (2019); Roberts et al. (2020); Lin et al. (2022), logical reasoning abilities Wei et al. (2022); Zhou et al. (2023a); Saparov et al. (2023), general problem-solving capabilities Kojima et al. (2022a) and more.

Recently there have also been studies on instruction-following - for instance, FoFo Xia et al. (2024) evaluates models on format-following tasks and studies the ability of LLMs to generate outputs in existing real-world formats. In a similar vein, IFEval Zhou et al. (2023b) assesses LLMs' ability to follow arbitrary task-specific instructions (e.g.) based on response length, casing, etc, focusing primarily on whether the instructions are followed rather than the correctness of the output for the task. InFoBench Qin et al. (2024) advances this research by introducing a metric known as the 'Decomposed Requirements Following Ratio' (DRFR) which is based on each aspect of an instruction that needs to be met. Along with 500 diverse instructions and 2,250 decomposed questions, InFoBench offers performance evaluation using OpenAI's GPT4, across multiple constraint categories and highlights key areas where advanced LLMs can improve in complex instruction-following tasks. LLMBar (Zeng et al., 2024) is another contribution to this area, as it provides a meta-evaluation benchmark specifically designed to test an LLM evaluator's ability to discern instruction-following outputs. The benchmark consists of 419 manually curated pairs of outputs, where one output adheres to instructions and the other, while potentially more engaging or deceptive, does not. Li et al. (2024) propose a method to evaluate instruction following ability via verbalizer manipulation. Specifically, they modify the classification task labels with different verbalizers which may or may not be semantically relevant to the task. They observe that all models fail to follow instructions when they instruct the model to flip the labels (unnatural setting). They evaluate the framework on mostly traditional NLP tasks like Sentiment Analysis, textual entailment etc.

Our work builds upon these efforts by developing a benchmark that allows for easy verification of both task performance and instruction-following capabilities simultaneously. We augment existing knowledge benchmarks by creating instructions that are *conditional* on answering the QA-based knowledge task correctly. We also include instructions that are applied on the candidate space of answers provided in these knowledge tasks. Our approach of applying instructions on knowledge tasks provides an easy way of measuring performance. Further, it also allows us to study the interactions between knowledge and instruction following, and to investigate whether instructions serve as distractors for the original knowledge task when the instructions should result in no change to the original answer of the knowledge task.

# 3 INSTRUCTION-FOLLOWING EVALUATION DATASET

We now describe the process for creating our evaluation dataset.

**Design Principles:** We develop our instructions keeping the following design principles in mind: (i) We would like instructions to be unambiguous and be presented in a way that can be communicated clearly - if humans cannot follow the instructions and agree on the same output, LLMs should and likely would not be able to. (ii) We would like them to be easy to follow and not require complex reasoning abilities to follow so that models at all scales have a fair chance of success, (iii) The instructions need to have deterministic outputs that use the original answers of the knowledge-task or the candidate space of answers, or both, so that they can be evaluated easily with instruction specific scorers. (iv) We would like our benchmark to be based on a diverse mix of knowledge tasks, and be easily extensible to new ones.

## 3.1 KNOWLEDGE AND REASONING TASKS

We select the following knowledge tasks that are commonly used in LLM evaluations as the basis for our instruction-following benchmark. These datasets involve either binary classification or multiple-choice-questions (MCQs) spanning different reasoning and problem-solving skills.

(i) **MMLUPro** (Wang et al., 2024): MMLUPro extends the MMLU dataset to make it more challenging by a) increasing the number of options from four to ten and b) increasing problem difficulty by focusing on more reasoning oriented problems. We consider all 14 subjects in the MMLUPro benchmark. We cap the maximum number of samples for each subject to be 150 samples.

(ii) **MathQA** (Amini et al., 2019): MathQA dataset consists of math word problems presented as Multiple-Choice-Questions (MCQs). Given a math question and four options, the model has to select the correct answer.

We also select a few common-sense and reasoning datasets:

(iii) **BoolQ** (Clark et al., 2019): BoolQ is a boolean question-answering dataset. Given a passage and a boolean question around the passage, the model has to select either *True* or *False*.

(iv) **PIQA** (Bisk et al., 2020): Physical Interaction: Question Answering (PIQA) involves answering questions that involve commonsense reasoning around physical objects. Given a question and two options, the model has to select the most plausible option.

(v) **Winogrande** (Sakaguchi et al., 2021): Winogrande involves a fill-in-the-blank task with binary options, the model has to select the correct option for a given sentence. The task involves reasoning for pronoun resolution.

We select a subset of 1500 samples randomly from each of the above datasets.

## 3.2 INSTRUCTION CATEGORIES

Unlike datasets that require open-ended generation for answering, our selected tasks have a structured answer-space. This allows us to craft instructions using these answer-spaces in a way that can be verified easily. We define the following instruction categories.

Table 2: Categories of instructions and the number of instances of each in the Full and Lite subsets.

| Instruction Group | Name | Definition | # Instances Full | Lite |
|---|---|---|---|---|
| String Manipulation | alternate_case_correct_answer | Print the text corresponding to the correction candidate answer of knowledge task in alternate case | 7867 | 950 |
| | capitalize_correct_answer | Print the text corresponding to the correct candidate answer of the knowledge task in upper case. | 7867 | 950 |
| | reverse_correct_answer_alternate_case | Reverse the text corresponding to the correct candidate answer of the knowledge task and print it in alternate case. | 9573 | 1383 |
| | reverse_correct_answer | Print the text corresponding to the correct answer in reverse | 7868 | 951 |
| Format Correct Answer | numformat_numeric_answer | Apply a specified decimal formatting the correct answer if it a is numeric quantity, otherwise print the correct answer as is. | 11336 | 1600 |
| | print_correct_answer_in_words | If the correct answer is a numeric quantity, display the numeric quantity in words, otherwise print the correct answer as is. | 9874 | 1320 |
| | print_correct_answer_append_string | Append a pre-specified string to the text associated with the correct candidate answer. | 7867 | 950 |
| Operations on List (Conditional on Correct Answer) | increment_incorrect_numeric_answers_by_one | If the candidate answer values are numeric quantities increment them by one and show them as a list. Other value types are not modified. | 7117 | 825 |
| | sort_only_incorrect_answers | Sort the candidate answers that are incorrect in ascending order | 7867 | 950 |
| | use_incorrect_options_to_create_string | Sort the incorrect candidates in ascending order and take the last character of the text associated with each incorrect option to create a string | 7868 | 951 |
| Operations on List | sort_options_to_create_string | Sort all candidate answers in ascending order and use the last character of the text associated with each incorrect candidate to create a string. | 7867 | 950 |
| Numeric Manipulation | increment_correct_numeric_answer_by_one | If the correct answer is a numeric quantity, increment it by one, otherwise print the correct answer as is. | 9757 | 1352 |

(i) **String Manipulation:** This operation involves manipulating the characters within the correct answer. We apply simple transformations like changing the case of the answer text or reversing the answer text, etc.

(ii) **Format Correct Answer:** This operation involves displaying the correct answer in the specified format. This involves printing any numeric answers in words or appending a string to the correct answer, etc.

(iii) **Numeric Manipulation:** This instruction involves incrementing a numeric quantity by one and has no effect on non-numeric answer text.

(iv) **Operations on list (Conditional):** This operation involves conditionally manipulating the candidate answer space – for instance, incrementing incorrect answers by one, sorting the incorrect answers, etc.

(v) **Operations on list:** These are simple instructions that do not depend on the correct answer of the original knowledge tasks. Examples, include - sorting all candidate options, concatenating characters from each candidate option, etc.

For each instruction category, we create multiple instructions. Table 2 presents the 13 instructions we have included in our work. The task prompts (instructions) for each of the 13 instruction types with an example are available in the Appendix (Section A.2).

**Instruction Creation:** To create each instruction, the authors iteratively refined them until all the authors had complete agreement in the output when they followed them manually. Examples of aspects of iterative improvement include - explicitly making clear what is not to be included in the output, how the output is to be presented, etc. We then asked 2 computer science researchers to follow and generate the output for 75 instructions across all our instruction types and datasets. We found that both the researchers were able to follow our instructions successfully and generated the same response for 93.33% of the instances. The first annotator generated the correct response for 98.67% of the instances, while the second annotator for 94.67% of the instances. Upon analyzing their responses, we found the only instruction-following error was rounding off the decimal number when truncating to two decimal places. We also found very few human errors in the annotator's response, specifically for instructions like *reverse_correct_answer_alternate_case* on datasets with long output text such as PIQA.

**Answering baseline-instructions:** We additionally develop two baseline instructions – (1) printing the correct answer option[1] from the candidate space (*print_correct_answer_label*), and (2) printing only the text associated with the correct answer option (*print_correct_answer_text*).

---

[1] We use 'label' and 'option' interchangeably to denote the candidates in a multiple-choice QA task.

**Instructions with no-effect**: Certain instructions may be inapplicable for some knowledge tasks. For example, in the MathQA dataset, some instances have *none of these* as the correct answer and are not numeric. Here, instructions such as $num\_format\_numeric\_answer$ or $increment\_correct\_numeric\_answer\_by\_one$ will not affect the existing answer of the knowledge-task. We refer to these instructions as "*distractor*" instances and expect that in these instances, models should perform as well as they do on the answering baseline-instructions. We include details and statistics of such instructions in the Appendix (Section A.4 Tables 7 and 9).

## 3.3 METRICS

**Exact Match:** We report the model performance as exact match under two settings - *strict* and *loose*. In the **strict** setting, we perform basic string parsing (removing beginning and ending whitespaces, quotations, etc.) and compare the model prediction to the expected output for the applied instruction.

However, we observe that models often make errors when following the primary instruction. These could be minor copying errors, such as missing a period or comma, or even fixing typos within the provided options. On the other hand, they could also be instruction following mistakes, where for instance the option label is added to the response even when the prompt explicitly states otherwise. Given that we do not expect models to make such mistakes given clear instructions, we use the strict metric in the majority of our evaluations.

However, we also define a relaxed version of the exact match called **loose** exact match, allowing for a Levenshtein distance Levenshtein (1966) of two edit operations between the prediction and ground truth. Additionally, we also perform whitespace-free matching as part of our loose criterion. Similar to Zhou et al. (2023b), we consider our loose match as a complement to the strict one.

## 3.4 BENCHMARK DATASET

We create two versions of our benchmark dataset - 'Full' and 'Lite' (for lower inference costs).

**Full Benchmark**: We select a subset of 1500 samples randomly from each datasets and apply each applicable instruction on the same. For MMLUPro, we consider a subset of 150 samples per subject and apply each applicable instruction.

**Lite Benchmark:** We select a subset of 150 samples randomly from the full version created above for each dataset and apply each applicable instruction on the same. For MMLUPro, we consider a subset of 25 samples per subject and apply each applicable instruction. Statistics for the above two versions are available in presented in Table 2 in the appendix. Detailed statistics for each dataset and the instruction types are provided in the appendix section A.4. Additionally, each benchmark includes a set of instances when instructions have no effect (called the no-effect subset).

### 3.4.1 BENCHMARK RANKING

An effective instruction-following model should not only be capable of following a variety of instructions across different knowledge-tasks but should also be unaffected by instructions when they are inapplicable i.e, they should be robust to 'distractors'. Therefore, we define an overall benchmark score for a model as its arithmetic mean of the following:

**Exact-Match Score** ($\mu_{EM}$): We compute the micro-average of the exact-match scores using all instances of every instruction type in the benchmark.

**Instruction Category Score (IC Score)**: We compute the micro-average exact-match scores for every instance per instruction category and then compute the arithmetic mean.

**Knowledge Task Subset Score: (KTS Score):** We compute the micro-average exact-match scores for every instance per knowledge-task, and then compute the arithmetic mean.

**Exact Match Score on 'Instructions with no-effect'** ($\mu'_{EM}$): We compute the micro-average of all instruction instances in the benchmark that have no effect on the original knowledge-task answers (i.e.) 'distractors'.

Table 3: List of Models evaluated on our benchmark.

| Small ($< 7B$ parameters) | Medium ($7 - 30B$ parameters) | Large ($> 30B$ parameters) | Frontier |
|---|---|---|---|
| Llama-3.2-1B-Instruct (1B) | Mistral-7B-Instruct-v0.3 (7B) | Qwen2.5-32B-Instruct (32B) | Llama-3.1-405B-Instruct (405B) |
| Qwen2.5-1.5B-Instruct (1.5B) | Qwen2.5-7B-Instruct (7B) | Llama-3.1-70B-Instruct (70B) | GPT-4o-mini-2024-07-18 |
| Llama-3.2-3B-Instruct (3B) | Phi-3-small-8k-instruct (7B) | Qwen2.5-72B-Instruct (72B) | GPT-4o-2024-08-06 |
| Qwen2.5-3B-Instruct (3.0B) | Llama-3.1-8B-Instruct (8B) | | |
| Phi-3.5-mini-instruct - (3.8B) | Gemma-2-9b-it (9B) | | |
| | Phi-3-medium-4k-instruct (14B) | | |
| | Qwen2.5-14B-Instruct (14B) | | |
| | Gemma-2-27b-it (27B) | | |

## 4 EVALUATION

We present an evaluation on our benchmark using a variety of models and study the following research questions: (i) Do models display a difference in performance on the two simple baseline instruction tasks? (ii) Do models display a variation in performance across our different instruction categories? (iii) Are models robust to, or get distracted by instructions that do not apply to the task? (iv) Does the size of a model impact its instruction-following capability?

### 4.1 MODELS AND INFERENCE

We evaluate our benchmark on a range of open instruction-tuned models and parameter sizes. For ease of presentation, we categorize them based on their parameter count as shown in Table 3. Our inference code uses vLLM Kwon et al. (2023) for running the evaluations. We use greedy decoding for generations and `bf16` as floating point precision. We generate a maximum of `1024` tokens per instance. We use A100 80GB GPUs for running inference. We use an instance hosted by a cloud provider for Llama-3.1-405B-Instruct, while we use OpenAI APIs for GPT4-o and GPT4-o-mini models.

In all our experiments, we perform zero-shot Chain-of-Thought (CoT) Wei et al. (2024) reasoning. Models see the same prompt based on prompt guides for the original knowledge tasks in lm-evaluation-harness framework Gao et al. (2024) and OpenAI evals.[2] We instruct the model to generate reasoning first and then the answer (See examples in Appendix Section A.2). We write custom post-processing scripts to extract the model's answer as described in the next section.

### 4.2 OUTPUT POST-PROCESSING

All our task prompt templates, as shown in Appendix A.2, explicitly instruct the model to provide their final response after a '`Response:`' keyword. As part of our *strict* evaluation metric (Section 3.3), we search for and extract the response after this keyword while computing the metrics.

However, we observe that models may not always follow this, and can instead generate a wide range of other keywords (e.g.) ⟨*the final answer is*, *the output is*, *etc*⟩, or no keyword at all. Given the diverse possible responses, we make a good-faith attempt to capture these patterns as part of our *loose* evaluation to classify a wider range of model responses. In our subsequent results, we use the loose evaluation for error analysis, and denote the specific type of strategy elsewhere.

### 4.3 RESULTS

We begin this section by first presenting our results on the answering baseline-instructions and then proceed to our results on instruction-following for the different categories. We then look at the impact of distractors and knowledge-task characteristics on model performance.

#### 4.3.1 PRINTING THE CORRECT ANSWER

We begin our experiments with the simplest task – given a multiple-choice question with option labels and their texts, we instruct the models to print the text associated with the correct answer instead of the answer label. From a knowledge perspective, this task is no harder than selecting the

---

[2]`https://github.com/openai/simple-evals`

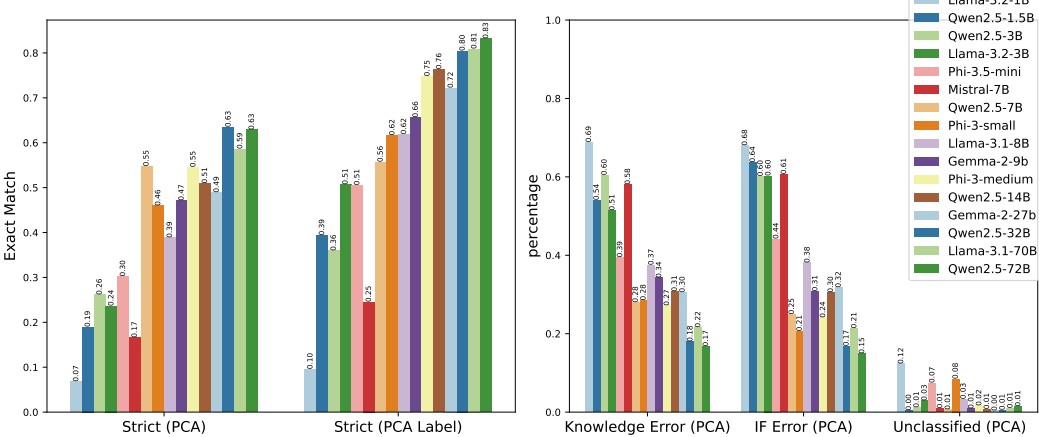

Figure 1: Left: Average exact match performance across all tasks for the *print_correct_answer* and *print_correct_answer_label* instructions. Right: Knowledge and instruction following (IF) errors across all tasks for the *print_correct_answer* instruction. A lower error is better. Both results shown using Full Benchmark data. Lite Benchmark results can be found in Appendix Figure 5.

answer label. However, as shown on the left in Figure 1, we observe a significant drop ($\sim 20\%$ on average) in knowledge-task performance when instructing the model to respond with the text associated with the answer instead of its label. The pattern is consistent for frontier models like *GPT-4o* on the Lite Benchmark (Figure 5).

We hypothesize that this drop in performance could, in part, be due to the training process resulting in models being over-fit to certain input/output task formats, resulting in worse instruction following for other formats. Some common issues we observed include models outright ignoring the instruction and continuing to generate labels, or generating only Chain-of-Thought reasoning without a final answer, missing the output keyword specified in the prompt, etc, reflected by the knowledge and instruction following errors in Figure 1 on the right.

We observe that the errors decrease as the inherent model capability and size increases. Note that incorrect answers could correspond to both knowledge and instruction following errors. The figure also shows that we capture most errors. Figures 26-28 show the error analysis for different model families, where we observe larger models making fewer errors ($20\% - 80\%$ reduction) for the Llama and Qwen models. The Phi model family however does not show this trend, calling for a closer look at their instruction training methodology. Figures 29-32 takes a deep-dive at the error distribution for each instruction category across model scales. We observe that models make the most errors on string manipulation tasks, and model scale does little to mitigate this. For the other categories, errors reduce as the model size increases. Inspired by this, and to further illustrate the challenges that LLMs face on simple instruction-following, we study their performance when the final output requires first inferring the correct answer, and then applying operations specified by the instructions on the correct answer.

### 4.3.2 ANSWER-CONDITIONED INSTRUCTION PERFORMANCE

We present results from different model scales across our five instruction categories in Figures 2a-2d. We compare this to their corresponding performance on the baseline task of *print_correct_answer* (PCA).

**Small-Scale and Medium-Scale models:** We observe a $10\% - 40\%$ drop in performance compared to the baseline across all instruction categories (Figures . 2a and 2b). In particular, we notice that all models struggle on our set of string and numeric manipulation instructions, suggesting a bias towards certain input/output format instructions (see Figure 24a and 24b for instruction-specific results for each model).

We further notice that models such as Qwen-2.5-14B and Llama-3.1-8B exhibit good loose evaluation scores for numeric manipulations and formatting the correct answer, but suffer a large drop

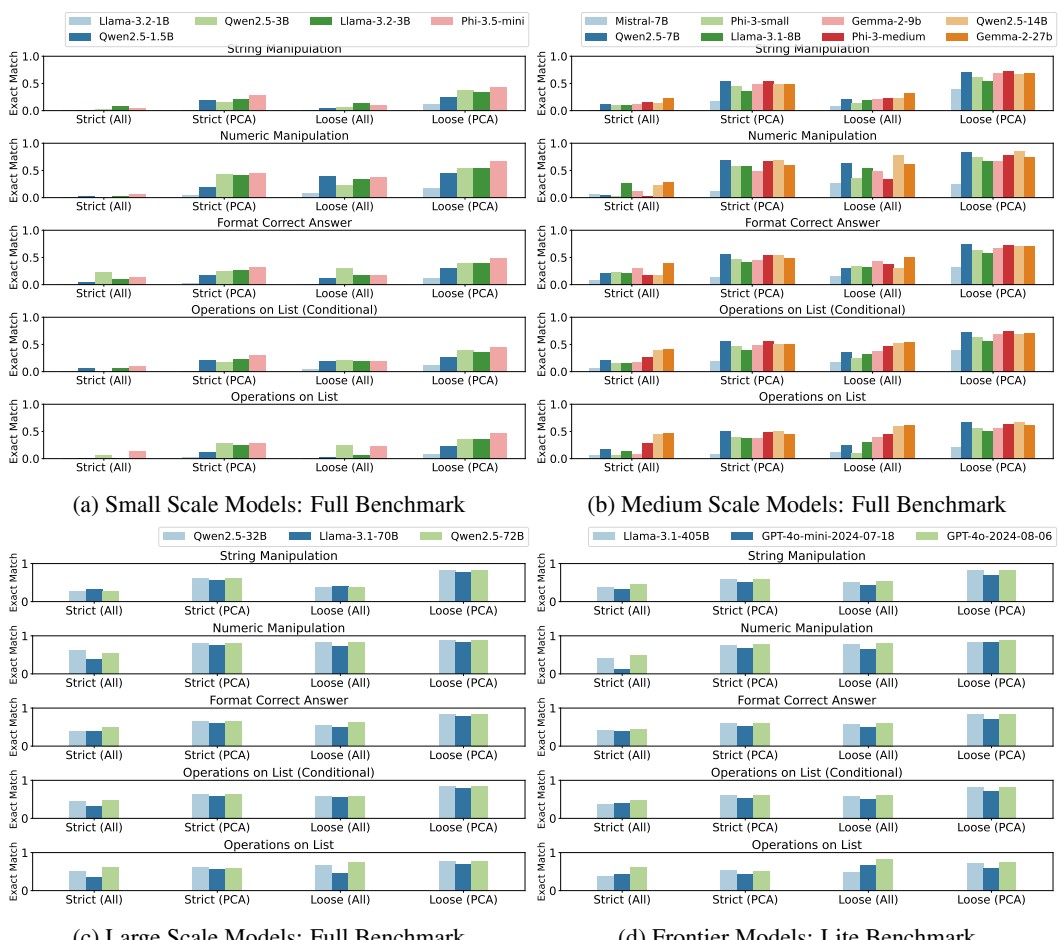

(a) Small Scale Models: Full Benchmark

(b) Medium Scale Models: Full Benchmark

(c) Large Scale Models: Full Benchmark

(d) Frontier Models: Lite Benchmark

Figure 2: Performance variation (strict and loose) of exact match scores across the different answer-conditioned instruction categories (All) from Table 2 compared to corresponding performance on $print\_correct\_answer$ (PCA).

in the corresponding strict evaluation. This difference suggests that these models are able to grasp the expectations from the instructions, but fail to follow them precisely. Examples of this include incrementing an answer by $0.1$ when asked to increment by one, or only returning the special string when instructed to append it to the correct answer text, or even adding/missing characters from the provided options when instructed to return them as is. Finally, all models also find the operations on list categories to be challenging – where interestingly the performance of models across both sets - conditional on correct answers vs. not, is similar.

**Large-scale and Frontier models:** The improved capabilities of these larger models are evident from the absolute improvement in performance as shown in Figures 2c and 2d. We also observe a smaller drop in performance between their loose evaluation and strict evaluation scores, reflecting more precise instruction following. However, the trend of performance deterioration ($5\% - 40\%$) across instructions, compared to their respective baseline knowledge-answering tasks, still persists in these models, demonstrating opportunities to improve their instruction following.

### 4.3.3 EFFECT OF PARAMETER SIZE WITHIN A MODEL FAMILY

We report the performance on Full Benchmark for models from the Llama family and Qwen family of models in Figures 33 and 34 in the appendix. We observe a consistent pattern of improvements in instruction following-ability with increase in model capacity for the Llama family. However, this is not the case for Qwen family of models. Specifically, for some instructions

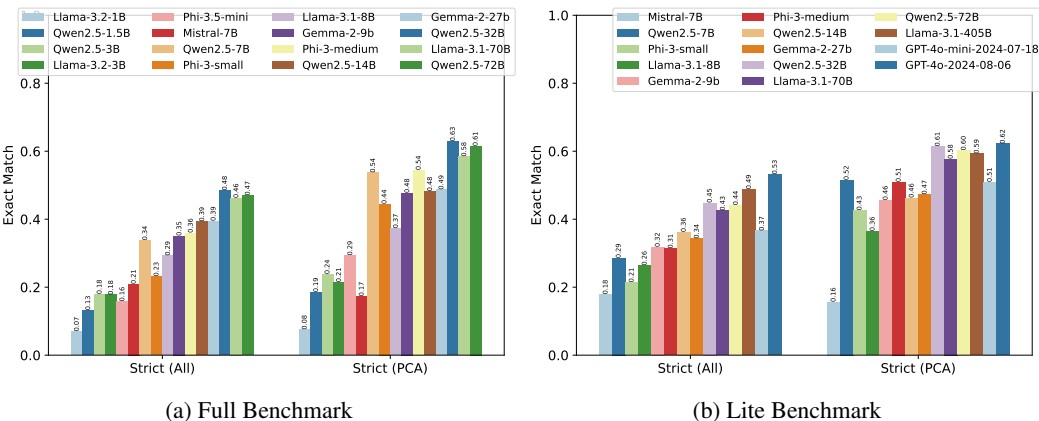

(a) Full Benchmark          (b) Lite Benchmark

Figure 3: Impact of distractor instructions on exact match performance across tasks and instructions, compared to its corresponding *print_correct_answer* performance. A drop indicates the model getting distracted by an inapplicable instruction.

like *print_correct_answer*, *print_correct_answer_label*, *sort_only_incorrect_answers* the Qwen 1.5B model outperforms 3B model. Qwen 3B model is better than Qwen 7B and 14B variants for the *print_correct_answer_append_string* instruction. We consistently see 32B and 72B variants outperforming other models by a significant margin.

### 4.3.4 INSTRUCTIONS AS DISTRACTORS

Our dataset also includes instructions that apply only when certain properties of a knowledge-task answer are fulfilled. For instance, instructions for incrementing the correct answer by one if numeric, formatting numeric values, and printing any numeric answers in words, do not apply on tasks with textual answers. They serve as distractors, and we expect model performance to be unaffected since these instructions are not applicable and do not alter the original knowledge-task answer.

However, from Figure 3, we observe that there is a 5-20% drop in small, medium, large, and frontier scale models. In figures 39a, 39b, 39c we report details of how different model families (Llama, Qwen and Phi) are affected by distractors, at different scales. We find that the Llama family and Phi of models are extremely distracted by instructions that require reversing and casing text (even though the instruction is inapplicable on numerical data), and report a drop of nearly 75-78% while Qwen family of models (at all scales) is relatively robust to such distractors. On the other hand, distractor instructions that are based on numeric operations lead to a minor drop in performance in Llama and Qwen models but still affect Phi family of models significantly. While model failures in the presence of distractors have been studied before (Shi et al., 2023; Feng et al., 2024), to the best of our knowledge this is the first work to study them in an instruction-*following* setting.

### 4.3.5 KNOWLEDGE-TASK CHARACTERISTICS AND INSTRUCTION-FOLLOWING

As seen in Figure 4, the performance drop for models for an instruction category can also be dependent on the nature of the knowledge-task. For instance, models appear to have a larger relative drop on MathQA as compared to MMLUPro for the numeric manipulation instruction category. Models also struggle more on string manipulation operations on PIQA - probably because of the long sentences that are part of answer candidates. Other knowledge-task and model scales have been presented in Appendix Section A.6.

### 4.4 BENCHMARK

We report the *strict* scores of the medium, large and frontier models on the Lite Benchmark in Table 4. Unsurprisingly, GPT4o model performs the best on our benchmark data while large and medium-scale models like Llama-3.1 405B, Qwen2.5 72B, and, Qwen2.5 32B models appear to be better than other openly available models including Llama-3.1-70B-instruct and the Gemma family of models. We also include the results on the full benchmark in Appendix Table 5. We note that the ranking of models is largely consistent and that small models are much weaker than larger models.

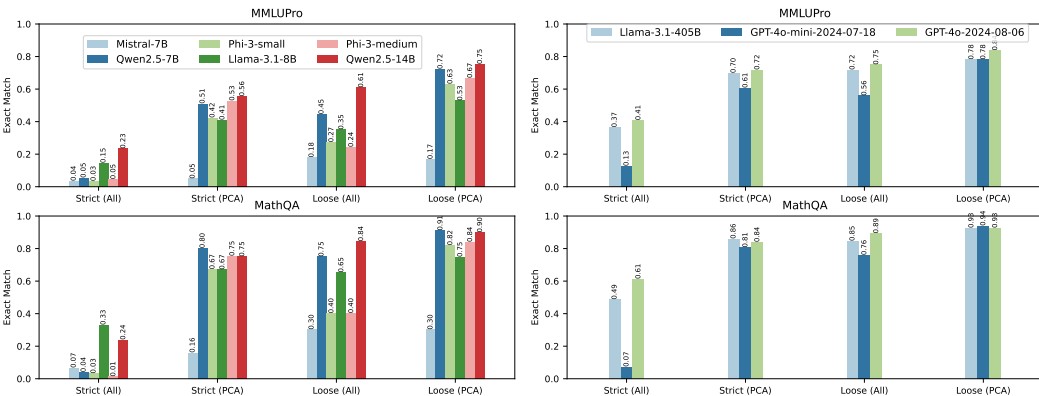

(a) Medium Scale Models: Numeric Manipulation          (b) Frontier Models: Numeric Manipulation

Figure 4: Performance variation (strict and loose) of exact match scores for the Numeric Manipulation instruction category compared to its corresponding performance on $print\_correct\_answer$ (PCA).

Table 4: Performance of the Medium, Large and Frontier Models on our Lite Benchmark - models ranked in order of performance using the average score (higher is better).

| Models | $\mu_{EM}$ | IC Score | KTS Score | $\mu'_{EM}$ | Average Score |
|---|---|---|---|---|---|
| GPT-4o-2024-08-06 | 0.4790 | 0.4990 | 0.5543 | 0.5318 | 0.5161 |
| Llama-3.1-405B | 0.4236 | 0.4537 | 0.4920 | 0.4883 | 0.4644 |
| Qwen2.5-72B | 0.4021 | 0.4690 | 0.4548 | 0.4410 | 0.4417 |
| Qwen2.5-32B | 0.3710 | 0.4402 | 0.4311 | 0.4481 | 0.4226 |
| Llama-3.1-70B | 0.3394 | 0.3832 | 0.3946 | 0.4253 | 0.3856 |
| GPT-4o-mini-2024-07-18 | 0.3601 | 0.3327 | 0.4299 | 0.3659 | 0.3722 |
| Gemma-2-27b | 0.3254 | 0.3673 | 0.3902 | 0.3430 | 0.3565 |
| Qwen2.5-14B | 0.2508 | 0.2996 | 0.2980 | 0.3620 | 0.3026 |
| Phi-3-medium | 0.2056 | 0.2250 | 0.2512 | 0.2932 | 0.2437 |
| Gemma-2-9b | 0.1716 | 0.1952 | 0.2133 | 0.3092 | 0.2223 |
| Qwen2.5-7B | 0.1700 | 0.1860 | 0.2029 | 0.2849 | 0.2109 |
| Llama-3.1-8B | 0.1568 | 0.1996 | 0.1840 | 0.2637 | 0.2010 |
| Phi-3-small | 0.1418 | 0.1535 | 0.1780 | 0.1970 | 0.1676 |
| Mistral-7B | 0.0566 | 0.0786 | 0.0755 | 0.1789 | 0.0974 |

## 5  DISCUSSION & CONCLUSION

In this work, we demonstrated how modern LLMs fail to follow simple instructions. We took a novel approach to studying instruction-following by grounding instructions on existing knowledge tasks. Our approach has the advantage of being easily extendable for new instruction types and domains, while also enabling LLM-free evaluations with some degree of automated error analysis. We demonstrated that not only do models fail to follow simple instructions (e.g.) printing the answer text instead of the label, but their performance drops further when compound but simple, instructions are included. Even when instructions that should have no effect on the knowledge-tasks are used, models at all scales report a drop in performance, though the extent of deterioration varies. As models are increasingly being viewed as agents and assistants, it is crucial that models have better guarantees of following user instructions. As our benchmark demonstrates, there is a lot of scope for improvement and we hope the community finds it helpful in improving the current state-of-the-art.

Lastly, before concluding, we would like to re-emphasize the choice of the strict measure to study performance - if instructions specify how the task is to be completed then models should not add extraneous text, respond by rephrasing the question as part of the final response, make copying errors, etc. The nature of errors made by models as reflected in the difference between loose and strict scores, the automated error analysis sets and the large amount of unclassified errors highlights that instruction-tuning of LLMs requires special focus on instruction-*following*.

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

## A  APPENDIX

We describe how we automatically classify errors in section A.1. We list all instructions with example input, ground truth, and expected instruction output in A.2. We the report results on the Full Benchmark in A.3. The detailed statistics of Full ad Lite Benchmark are presented in A.4. Section A.5 presents the comparison between model's performance on print correct answer and print correct answer labels tasks on the Lite Benchmark. Section A.6 presents performance of different models for each instruction category in comparison with its corresponding performance on $print\_correct\_answer$ (PCA).

### A.1  AUTOMATED CLASSIFICATION OF ERRORS

For each instruction in the 'String Manipulation', 'Format Correct Answer', and 'Numeric Manipulation' instruction categories, we create a set of error classes that are based on the incorrect answers to the original knowledge task and the subsequent application of the instruction on that incorrect answer. We create 'instruction-following' error sets (IFError) and 'knowledge-error sets' (KnowledgeError) as follows:

**IFError**: We inspected some representative model outputs and created a set of errors that LLMs could plausibly make for each instruction. This set includes errors such as: (i) answering with an answer label (option) when requiring/using the text, (ii) answering with any candidate answer text without applying the instruction, (iii) a combination of these, (iv) instruction specific errors based on these – for example, for the instruction $alternate\_case\_correct\_answer$ such error cases could include alternate casing starting with a lowercase character (the instruction text explicitly requires that it start with upper case).

**KnowledgeError:** For every instruction instance, this set includes: (i) Answering with the incorrect candidate answer – regardless of whether it is its label or text, (ii) Applying the instruction (when applicable) on the incorrect candidate answer.

Note that these error sets are not mutually exclusive – a model response to the alternate casing instruction that starts with lowercase and is applied on the incorrect answer candidate is an 'IFError' as well as a 'KnowledgeError'. Further, these error sets are meant to be *high-precision* to aid analysis and cannot reasonably expected to anticipate and cover everything LLMs might respond with. Hence, there will be model errors that we cannot classify and we intend to continue to expand the

benchmark's error classification capabilities. We would also like to highlight that if models make a lot of errors that are not classified, it is likely that those are more severe instruction-following errors.

## A.2 PROMPTS

We now list the prompts for each instruction and dataset combination defined in our benchmark.

---

Dataset: Winogrande
Instruction: print_correct_answer_append_string

Given a sentence with a blank and 2 options namely 'A' and 'B', print the text associated with the option label that answers the question correctly. Do not print the option label. Without including any additional characters, also append the following text string at the end of the output:'ENDOFRESPONSE'. Do not add any special characters including quotations, spaces, new lines etc in the response. Think step by step and in the end, finish your response with 'Response:$RESPONSE' where $RESPONSE (without quotes) is the final output expected.
Sentence: It is an article of faith that the paper is more important than the exam , even though the _ weighs less heavily on the grade.
**Options:**
A. paper
B. exam
**Ground Truth:** A
**Instruction Output:** Response:paperENDOFRESPONSE

---

Dataset: Winogrande
Instruction: alternate_case_correct_answer

Given a sentence with a blank and 2 options namely 'A' and 'B', answer the question by printing the text associated with the correct option label in alternate case. The first character should be in uppercase and the following characters should alternate between lowercase and uppercase. Do not print the option label. Think step by step and in the end, finish your response with 'Response:$RESPONSE' where $RESPONSE (without quotes) is the final output expected.
Sentence: The wooden doors at my friends work are worse than the wooden desks at my work, because the _ material is stronger.
**Options:**
A. doors
B. desks
**Ground Truth:** B
**Instruction Output:** Response:DeSkS

---

**Dataset:** MathQA
**Instruction:** numformat_numeric_answer

Given a mathematical question and 5 options namely 'a', 'b', 'c', 'd', and, 'e', as candidate answers, print the text associated with the option label that answers the question correctly. If the answer is numeric print it in two decimal places as long as it contains no other string or units of measurement. Do not print the option label. Think step by step and in the end, finish your response with 'Response:$RESPONSE' where $RESPONSE (without quotes) is the final output expected.
**Question:** a man walking at the rate of 5 km / hr crosses a bridge in 15 minutes . the length of the bridge ( in meters ) is :
**Options:**
a. 600
b. 750
c. 1000
d. 1250
e. none of these
**Ground Truth:** d
**Instruction Output:** Response:1250.00

**Dataset:** MathQA
**Instruction:** sort_options_to_create_string

Given a mathematical question and 5 options namely 'a', 'b', 'c', 'd', and, 'e', as candidate answers, sort the list of options using their values, in alphabetical order. Use only the text associated with the option labels and not the option labels while sorting. Then, create a string by concatenating the last character of the text associated with each option value. If the last character is a special character (such as period, comma, quotation, etc) use the previous character. Print only the final string and not the sorted list. Think step by step and in the end, finish your response with 'Response:$RESPONSE' where $RESPONSE (without quotes) is the final output expected.
**Question:** marts income is 50 percent more than tims income and tims income is 40 percent less than juans income . what percentage of juans income is marts income
**Options:**
a. 124 %
b. 120 %
c. 96 %
d. 90 %
e. 64 %
**Ground Truth:** d
**Instruction Output:** Response:40604

**Dataset:** PIQA
**Instruction:** reverse_correct_answer

Given a question and two answer candidates 'A' and 'B', answer the question by printing the text associated with the correct option label, in reverse. Do not print the option label. Think step by step and in the end, finish your response with 'Response:$RESPONSE' where $RESPONSE (without quotes) is the final output expected.
**Question:** Butcher Shop
**Options:**
A. will decimate fish from the ocean into digestible pieces
B. will decimate a full cow into digestible pieces
**Ground Truth:** B
**Instruction Output:** Response:seceip elbitsegid otni woc lluf a etamiced lliw

Dataset: PIQA
Instruction: print_correct_answer

Given a question and two answer candidates 'A' and 'B', answer the question by selecting the value associated with the option label corresponding to the correct answer. Do not print the option label. Think step by step and in the end, finish your response with 'Response:$RESPONSE' where $RESPONSE (without quotes) is the final output expected.
**Question:** how to avoid paint spill when adding paint to your brush
**Options:**
A. Put a rubber band on your paint can to get rid of that excess glue on your paint brush, this will prevent spilling paint on the paint stir stick where the lid is.
B. Put a rubber band on your paint can to get rid of that excess glue on your paint brush, this will prevent spilling paint on the edge where the lid is.
**Ground Truth:** B
**Instruction Output:** Response:Put a rubber band on your paint can to get rid of that excess glue on your paint brush, this will prevent spilling paint on the edge where the lid is.

Dataset: Winogrande
Instruction: use_incorrect_options_to_create_string

Given a sentence with a blank and 2 options namely 'A' and 'B', create a string by concatenating the last character of every option value, excluding the option value corresponding to the correct answer. Use only the text associated with the option labels and not the option labels while sorting. If the last character is a special character (such as period, comma, quotation, etc) use the previous character. Think step by step and in the end, finish your response with 'Response:$RESPONSE' where $RESPONSE (without quotes) is the final output expected.
Sentence: Fiction books were interesting and easy to read for Logan but not Brett because _ enjoyed real tales.
**Options:**
A. Logan
B. Brett
**Ground Truth:** B
**Instruction Output:** Response:n

Dataset: PIQA
Instruction: reverse_correct_answer_alternate_case

Given a question and two answer candidates 'A' and 'B', reverse the text associated with the answer label that correctly answers the question. Print this reversed text in alternate case starting with upper case. Do not print the option label. Think step by step and in the end, finish your response with 'Response:$RESPONSE' where $RESPONSE (without quotes) is the final output expected.
**Question:** What else should I add to a peanut butter sandwich?
**Options:**
A. Take some pasta sauce and put it onto the other piece of bread with a knife.
B. Take some jelly and put it onto the other piece of bread with a knife.
**Ground Truth:** B
**Instruction Output:** Response:.EfInK A HtIw dAeRb fO EcEiP ReHtO EhT OtNo tI TuP DnA YlLeJ EmOs eKaT

Dataset: Winogrande
Instruction: use_options_to_create_string

Given a sentence with a blank and 2 options namely 'A' and 'B', create a string by concatenating the last character of every option value (not option label). If the last character is a special character (such as period, comma, quotation, etc) use the previous character. Think step by step and in the end, finish your response with 'Response:$RESPONSE' where $RESPONSE (without quotes) is the final output expected.
Sentence: Megan focused less on proper posture than Lindsey because _ wanted to become a model.
**Options:**
A. Megan
B. Lindsey
**Ground Truth:** B
**Instruction Output:** Response:ny

Dataset: MathQA
Instruction: print_correct_answer_label

Given a mathematical question and 5 options namely 'a', 'b', 'c', 'd', and, 'e', as candidate answers, answer the question by selecting the option label corresponding to the correct answer. Do not include the text associated with the option label in the answer. Think step by step and in the end, finish your response with 'Response:$RESPONSE' where $RESPONSE (without quotes) is the final output expected.
**Question:** a reduction of 20 % in the price of salt enables a lady to obtain 2 kgs more for rs . 100 , find the original price per kg ?
**Options:**
a. 12.6
b. 12.1
c. 12.5
d. 12.4
e. 12.7
**Ground Truth:** c
**Instruction Output:** Response:c

Dataset: PIQA
Instruction: increment_correct_numeric_answer_by_one

Given a question and two answer candidates 'A' and 'B', print the text associated with the option label that answers the question correctly. Note that if the correct answer is a numeric quanity, including dollar values and percentages but contains no other string or units of measurement, print the value after increasing its value by 1. Dollar values should be prefixed with '$'. Do not print the option label. Think step by step and in the end, finish your response with 'Response:$RESPONSE' where $RESPONSE (without quotes) is the final output expected.
**Question:** how to winterize windows
**Options:**
A. put weather stripping around them to stop air from escaping and air from coming in
B. put weather stripping around them to stop air from escaping and air from coming into the dishwasher
**Ground Truth:** A
**Instruction Output:** Response:put weather stripping around them to stop air from escaping and air from coming in

Dataset: MathQA
Instruction: sort_only_incorrect_answers

Given a mathematical question and 5 options namely 'a', 'b', 'c', 'd', and, 'e', as candidate answers, excluding the option that answers the question correctly, print a sorted list (ascending order) of the incorrect options. Do not print the option labels. Use the text associated with the option labels and not the option labels while sorting and printing. Think step by step and in the end, finish your response with 'Response:$RESPONSE' where $RESPONSE (without quotes) is the final output expected.
**Question:** the sector of a circle has radius of 21 cm and central angle 108 o . find its perimeter ?
**Options:**
a. 81.6 cm
b. 85.9 cm
c. 90 cm
d. 92 cm
e. 95 cm
**Ground Truth:** a
**Instruction Output:** Response:['85.9 cm', '90 cm', '92 cm', '95 cm']

Dataset: PIQA
Instruction: print_correct_answer_in_words

Given a question and two answer candidates 'A' and 'B', print the text associated with the option label that answers the question correctly. However, if the correct answer is a numeric value with no additional text (including percentages, currency, units of measurement etc), print the numeric answer in words. For example, if the answer is '32' print 'thirty-two' without quotes. Do not print the option label. Think step by step and in the end, finish your response with 'Response:$RESPONSE' where $RESPONSE (without quotes) is the final output expected.
**Question:** How do I make the pattern for the baby leather shoes?
**Options:**
A. Create a template on a piece of paper by placing your babies shoe on the paper and drawing around it.
B. Create a template on a piece of paper by placing your babies foot on the paper and drawing around it.
**Ground Truth:** A
**Instruction Output:** Response:Create a template on a piece of paper by placing your babies shoe on the paper and drawing around it.

> **Dataset: BoolQ**
> **Instruction: increment_incorrect_numeric_answers_by_one**
>
> Given a passage and a boolean question, and the possible answer candidates 'A' or 'B', print the list of incorrect answers (not the answer label). Increase each value by 1 while printing if it is a numeric quanity including dollar values, percentages but contains no other string or units of measurement. Do not print the option labels. Think step by step and in the end, finish your response with 'Response:$RESPONSE' where $RESPONSE (without quotes) is the final output expected.
> Passage: A Star Is Born is an upcoming American musical romantic drama film produced and directed by Bradley Cooper, in his directorial debut. Cooper also wrote the screenplay with Will Fetters and Eric Roth. A remake of the 1937 film of the same name, it stars Cooper, Lady Gaga, Andrew Dice Clay, Dave Chappelle, and Sam Elliott, and follows a hard-drinking country musician (Cooper) who discovers and falls in love with a young singer (Gaga). It marks the third remake of the original 1937 film (which featured Janet Gaynor and Fredric March), which was adapted into a 1954 musical (starring Judy Garland and James Mason) and then remade as a 1976 rock musical with Barbra Streisand and Kris Kristofferson.
> **Question:** is bradley cooper a star is born a remake
> **Options:**
> A. True
> B. False
> **Ground Truth:** A
> **Instruction Output:** Response:['False']

> **Dataset: PIQA**
> **Instruction: capitalize_correct_answer**
>
> Given a question and two answer candidates 'A' and 'B', answer the question by printing the text associated with the correct option label in uppercase. Do not print the option label. Think step by step and in the end, finish your response with 'Response:$RESPONSE' where $RESPONSE (without quotes) is the final output expected.
> **Question:** wool
> **Options:**
> A. can be used to line cookie tins
> B. can be used to line pants
> **Ground Truth:** B
> **Instruction Output:** Response:CAN BE USED TO LINE PANTS

## A.3 FULL BENCHMARK LEADERBOARD

We report *strict* scores of the small, medium, and, large models on the Full Benchmark in Table 5. We observe that Qwen2.5 72B and 32B variants outperforms all other models. Llama-3.1 70B is ranked third with a significant gap between the second best model Qwen2.5 32B. There is a significant drop in performance as the parameter size decreases.

## A.4 ADDITIONAL BENCHMARK STATISTICS

The following sections reports detailed statistics for the Full and Lite Benchmark. We report statistics for both instruction following and Instructions with no-effect subsets. We observe that for some dataset (knowledge tasks) and instruction combinations, the corresponding entries are zero indicating that there is no single instance where the instruction gets applied (Instructions with no-effect) or there is no single instance where the instruction doesn't get applied (instruction follow subset).

Table 5: Performance of the Small, Medium, and Large Models on our Full Benchmark - models ranked in order of performance using the average score (higher is better).

| Models | $\mu_{EM}$ | IC Score | KTS Score | $\mu'_{EM}$ | Average Score |
|---|---|---|---|---|---|
| Qwen2.5-72B | 0.4157 | 0.4827 | 0.4343 | 0.4697 | 0.4506 |
| Qwen2.5-32B | 0.3822 | 0.4501 | 0.4108 | 0.4846 | 0.4319 |
| Llama-3.1-70B | 0.3389 | 0.3532 | 0.3582 | 0.4623 | 0.3781 |
| Gemma-2-27b | 0.3392 | 0.3589 | 0.3727 | 0.3947 | 0.3664 |
| Qwen2.5-14B | 0.2545 | 0.2803 | 0.2767 | 0.3932 | 0.3012 |
| Phi-3-medium | 0.2013 | 0.1835 | 0.2167 | 0.3406 | 0.2355 |
| Gemma-2-9b | 0.1633 | 0.1583 | 0.1850 | 0.3409 | 0.2119 |
| Qwen2.5-7B | 0.1658 | 0.1515 | 0.1753 | 0.3366 | 0.2073 |
| Llama-3.1-8B | 0.1460 | 0.1741 | 0.1573 | 0.2927 | 0.1925 |
| Phi-3-small | 0.1352 | 0.1171 | 0.1568 | 0.2129 | 0.1555 |
| Phi-3.5-mini | 0.0883 | 0.0967 | 0.0932 | 0.1397 | 0.1045 |
| Llama-3.2-3B | 0.0704 | 0.0571 | 0.0780 | 0.1632 | 0.0922 |
| Mistral-7B | 0.0458 | 0.0541 | 0.0543 | 0.2093 | 0.0909 |
| Qwen2.5-3B | 0.0663 | 0.0704 | 0.0725 | 0.1090 | 0.0796 |
| Qwen2.5-1.5B | 0.0382 | 0.0347 | 0.0436 | 0.1223 | 0.0597 |
| Llama-3.2-1B | 0.0039 | 0.0037 | 0.0041 | 0.0184 | 0.0075 |

Table 6: Full Benchmark: Instruct Follow Stats

| | BoolQ | Physics | Health | Economics | Law | Philosophy | Business | Other | Chemistry | Psychology | History | Computer Science | Biology | Math | Engineering | PIQA | MathQA | Winogrande |
|---|---|---|---|---|---|---|---|---|---|---|---|---|---|---|---|---|---|---|
| numformat_numeric_answer | 0 | 150 | 53 | 150 | 150 | 44 | 150 | 150 | 150 | 150 | 47 | 150 | 150 | 150 | 150 | 553 | 1500 | 0 |
| increment_incorrect_numeric_answers_by_one | 1500 | 150 | 150 | 150 | 0 | 0 | 150 | 150 | 150 | 0 | 0 | 150 | 0 | 150 | 150 | 1500 | 1500 | 1267 |
| sort_only_incorrect_answers | 1500 | 150 | 150 | 150 | 150 | 150 | 150 | 150 | 150 | 150 | 150 | 150 | 150 | 150 | 150 | 1500 | 1500 | 1267 |
| use_options_to_create_string | 1500 | 150 | 150 | 150 | 150 | 150 | 150 | 150 | 150 | 150 | 150 | 150 | 150 | 150 | 150 | 1500 | 1500 | 1267 |
| print_correct_answer_label | 1500 | 150 | 150 | 150 | 150 | 150 | 150 | 150 | 150 | 150 | 150 | 150 | 150 | 150 | 150 | 1500 | 1500 | 1267 |
| print_correct_answer | 1500 | 150 | 150 | 150 | 150 | 150 | 150 | 150 | 150 | 150 | 150 | 150 | 150 | 150 | 150 | 1500 | 1500 | 1267 |
| reverse_correct_answer_alternate_case | 1500 | 150 | 150 | 150 | 150 | 150 | 150 | 150 | 150 | 150 | 150 | 150 | 150 | 150 | 150 | 1500 | 540 | 1267 |
| increment_correct_numeric_answer_by_one | 0 | 150 | 22 | 150 | 1 | 4 | 150 | 150 | 129 | 11 | 1 | 86 | 13 | 150 | 53 | 1500 | 1500 | 0 |
| alternate_case_correct_answer | 1500 | 150 | 150 | 150 | 150 | 150 | 150 | 150 | 150 | 150 | 150 | 150 | 150 | 150 | 150 | 1500 | 1500 | 1267 |
| print_correct_answer_append_string | 1500 | 150 | 150 | 150 | 0 | 150 | 150 | 150 | 150 | 150 | 150 | 150 | 150 | 150 | 150 | 1500 | 1500 | 1267 |
| print_correct_answer_in_words | 0 | 142 | 14 | 10 | 0 | 3 | 81 | 31 | 108 | 10 | 0 | 84 | 8 | 150 | 35 | 0 | 1500 | 0 |
| sort_options_to_create_string | 1500 | 150 | 150 | 150 | 150 | 150 | 150 | 150 | 150 | 150 | 150 | 150 | 150 | 150 | 150 | 1500 | 1500 | 1267 |
| reverse_correct_answer | 1500 | 150 | 150 | 150 | 150 | 150 | 150 | 150 | 150 | 150 | 150 | 150 | 150 | 150 | 150 | 1500 | 1500 | 1267 |
| use_incorrect_options_to_create_string | 1500 | 150 | 150 | 150 | 150 | 150 | 150 | 150 | 150 | 150 | 150 | 150 | 150 | 150 | 150 | 1500 | 1500 | 1267 |
| capitalize_correct_answer | 1500 | 150 | 150 | 150 | 150 | 150 | 150 | 150 | 150 | 150 | 150 | 150 | 150 | 150 | 150 | 1500 | 1500 | 1267 |

Table 7: Full Benchmark: Instructions with no-effect

| | Health | Economics | Math | Psychology | Law | Computer Science | Engineering | Biology | History | Philosophy | Chemistry | Other | Business | Physics | MathQA | PIQA | BoolQ | Winogrande |
|---|---|---|---|---|---|---|---|---|---|---|---|---|---|---|---|---|---|---|
| numformat_numeric_answer | 150 | 150 | 150 | 150 | 150 | 150 | 150 | 150 | 150 | 150 | 150 | 150 | 150 | 150 | 1337 | 1285 | 1500 | 1267 |
| print_correct_answer_in_words | 150 | 150 | 150 | 150 | 150 | 150 | 150 | 150 | 150 | 150 | 150 | 150 | 150 | 150 | 1331 | 1500 | 1500 | 1267 |
| increment_correct_numeric_answer_by_one | 150 | 150 | 150 | 13 | 3 | 122 | 150 | 33 | 2 | 15 | 150 | 150 | 150 | 150 | 928 | 1500 | 1500 | 1267 |
| reverse_correct_answer_alternate_case | 30 | 48 | 0 | 0 | 0 | 0 | 0 | 0 | 0 | 0 | 0 | 0 | 0 | 0 | 1500 | 0 | 0 | 0 |
| reverse_correct_answer | 0 | 0 | 0 | 0 | 0 | 0 | 0 | 0 | 0 | 0 | 0 | 0 | 0 | 0 | 1 | 0 | 1 | 0 |
| use_incorrect_options_to_create_string | 0 | 0 | 0 | 0 | 0 | 0 | 0 | 0 | 0 | 0 | 0 | 0 | 0 | 0 | 0 | 1 | 0 | 0 |

Table 8: Lite Benchmark: Instruct Follow Stats

| | | | | | | | MMLUPro | | | | | | | | | | | |
|---|---|---|---|---|---|---|---|---|---|---|---|---|---|---|---|---|---|---|
| | BoolQ | chemistry | law | other | physics | math | biology | philosophy | psychology | economics | history | health | law | engineering | business | computer science | PIQA | MathQA | Winogrande |
| print_correct_answer | 150 | 66 | 41 | 82 | 65 | 53 | 53 | 50 | 54 | 69 | 49 | 60 | 38 | 74 | 73 | 54 | 253 | 352 | 150 |
| print_correct_answer_label | 150 | 66 | 41 | 82 | 65 | 53 | 53 | 50 | 54 | 69 | 49 | 60 | 38 | 74 | 73 | 54 | 253 | 352 | 150 |
| increment_correct_numeric_answer_by_one | 0 | 25 | 3 | 25 | 25 | 25 | 13 | 4 | 11 | 25 | 1 | 22 | 1 | 25 | 25 | 25 | 0 | 150 | 0 |
| sort_options_to_create_string | 150 | 25 | 25 | 25 | 25 | 25 | 25 | 25 | 25 | 25 | 25 | 25 | 25 | 25 | 25 | 25 | 150 | 150 | 150 |
| print_correct_answer_in_words | 0 | 25 | 25 | 25 | 25 | 25 | 8 | 3 | 10 | 10 | 0 | 14 | 0 | 25 | 25 | 25 | 0 | 150 | 0 |
| reverse_correct_answer | 150 | 25 | 25 | 25 | 25 | 25 | 25 | 25 | 25 | 25 | 25 | 25 | 25 | 25 | 25 | 25 | 150 | 150 | 150 |
| use_incorrect_options_to_create_string | 150 | 25 | 25 | 25 | 25 | 25 | 25 | 25 | 25 | 25 | 25 | 25 | 25 | 25 | 25 | 25 | 150 | 150 | 150 |
| use_options_to_create_string | 150 | 25 | 25 | 25 | 25 | 25 | 25 | 25 | 25 | 25 | 25 | 25 | 25 | 25 | 25 | 25 | 150 | 150 | 150 |
| print_correct_answer_append_string | 150 | 25 | 25 | 25 | 25 | 25 | 25 | 25 | 25 | 25 | 25 | 25 | 25 | 25 | 25 | 25 | 150 | 150 | 150 |
| increment_incorrect_numeric_answers_by_one | 150 | 25 | 25 | 25 | 25 | 25 | 0 | 0 | 0 | 25 | 0 | 25 | 0 | 25 | 25 | 25 | 150 | 150 | 150 |
| sort_only_incorrect_answers | 150 | 25 | 25 | 25 | 25 | 25 | 25 | 25 | 25 | 25 | 25 | 25 | 25 | 25 | 25 | 25 | 150 | 150 | 150 |
| numformal_numeric_answer | 0 | 25 | 25 | 25 | 25 | 25 | 25 | 25 | 25 | 25 | 25 | 25 | 25 | 25 | 25 | 25 | 150 | 150 | 0 |
| reverse_correct_answer_alternate_case | 150 | 25 | 25 | 25 | 25 | 25 | 25 | 25 | 25 | 25 | 25 | 25 | 25 | 25 | 25 | 25 | 150 | 150 | 150 |
| alternate_case_correct_answer | 150 | 25 | 25 | 25 | 25 | 25 | 25 | 25 | 25 | 25 | 25 | 25 | 25 | 25 | 25 | 25 | 150 | 150 | 150 |
| capitalize_correct_answer | 150 | 25 | 25 | 25 | 25 | 25 | 25 | 25 | 25 | 25 | 25 | 25 | 25 | 25 | 25 | 25 | 150 | 150 | 150 |

Table 9: Lite Benchmark: Instructions with no-effect

| | | | | | MMLUPro | | | | | | | | | | | | |
|---|---|---|---|---|---|---|---|---|---|---|---|---|---|---|---|---|---|---|
| | biology | health | law | engineering | chemistry | math | business | physics | psychology | history | other | computer science | economics | philosophy | MathQA | Winogrande | BoolQ | PIQA |
| print_correct_answer | 57 | 47 | 41 | 56 | 46 | 51 | 48 | 52 | 44 | 29 | 50 | 63 | 57 | 43 | 293 | 150 | 150 | 198 |
| print_correct_answer_label | 57 | 47 | 41 | 56 | 46 | 51 | 48 | 52 | 44 | 29 | 50 | 63 | 57 | 43 | 293 | 150 | 150 | 198 |
| reverse_correct_answer_alternate_case | 25 | 25 | 3 | 25 | 25 | 25 | 25 | 25 | 13 | 2 | 25 | 25 | 25 | 15 | 150 | 0 | 0 | 0 |
| print_correct_answer_in_words | 25 | 25 | 25 | 25 | 25 | 25 | 25 | 25 | 25 | 25 | 25 | 25 | 25 | 25 | 150 | 150 | 150 | 150 |
| increment_correct_numeric_answer_by_one | 25 | 25 | 25 | 25 | 25 | 25 | 25 | 25 | 25 | 25 | 25 | 25 | 25 | 25 | 150 | 150 | 150 | 150 |
| numformal_numeric_answer | 25 | 25 | 25 | 25 | 25 | 25 | 25 | 25 | 25 | 25 | 25 | 25 | 25 | 25 | 150 | 150 | 150 | 150 |
| reverse_correct_answer | 0 | 0 | 0 | 0 | 0 | 0 | 0 | 0 | 0 | 0 | 0 | 1 | 0 | 0 | 1 | 0 | 0 | 0 |
| use_incorrect_options_to_create_string | 0 | 0 | 0 | 0 | 0 | 0 | 0 | 0 | 0 | 0 | 0 | 0 | 0 | 0 | 0 | 0 | 0 | 1 |

## A.5 PRINTING THE CORRECT ANSWER

We present the comparison between model's performance on print correct answer and print correct answer labels tasks on the Lite Benchmark in Table 5. We observe that all models show a drop in performance when instructed to print correct answer instead of the label.

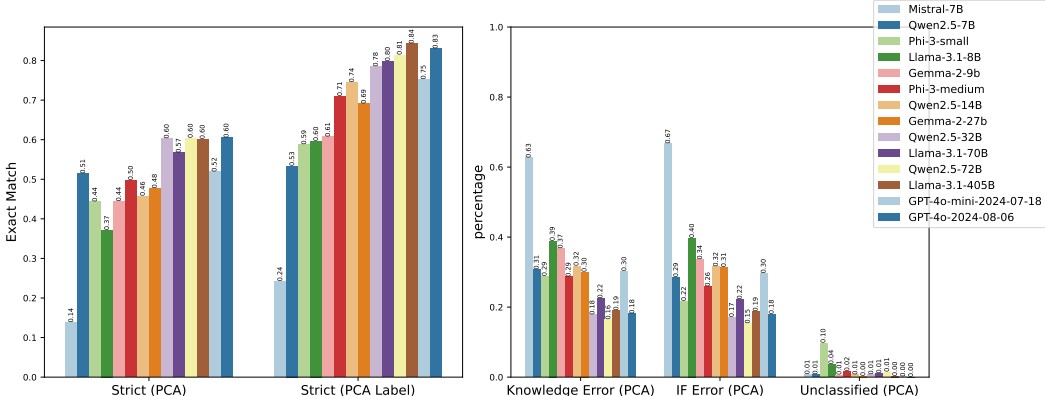

Figure 5: Lite Benchmark: Performance of LLMs on Printing the correct answer task and error comparison. PCA refers to *print_correct_answer* instruction and PCA label refers to *print_correct_answer_label*.

## A.6 KNOWLEDGE-TASK CHARACTERISTICS AND INSTRUCTION-FOLLOWING

We now present performance of different models for each instruction category in comparison with its corresponding performance on *print_correct_answer* (PCA). The patterns remains consistent.

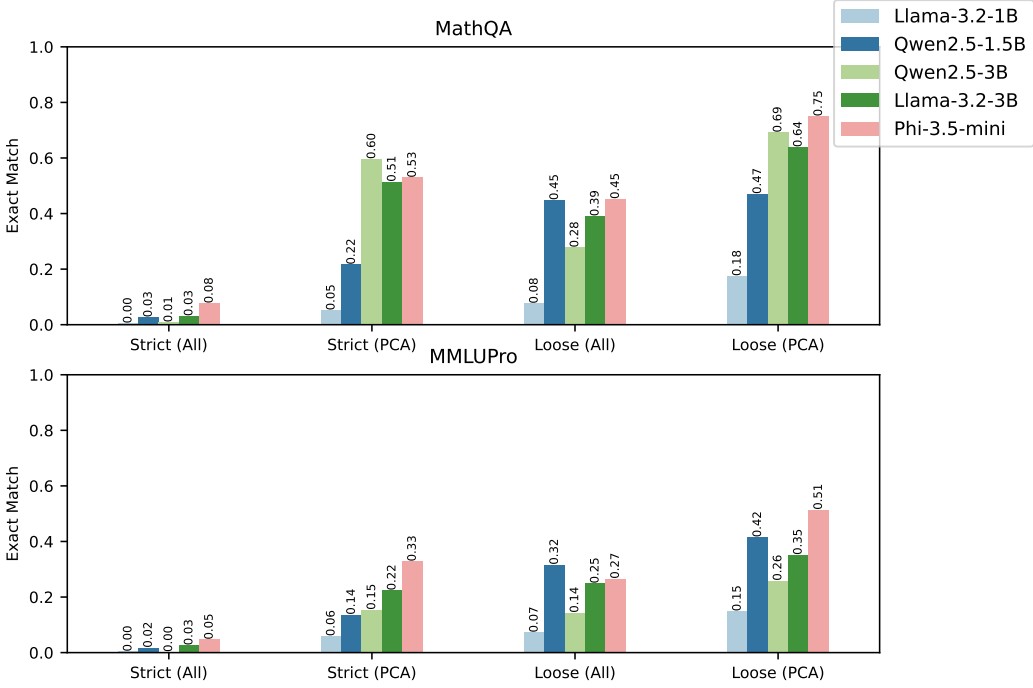

Figure 6: Small Scale Models: Performance variation (strict and loose) of exact match scores for the Numeric Manipulation instruction category compared to its corresponding performance on *print_correct_answer* (PCA).

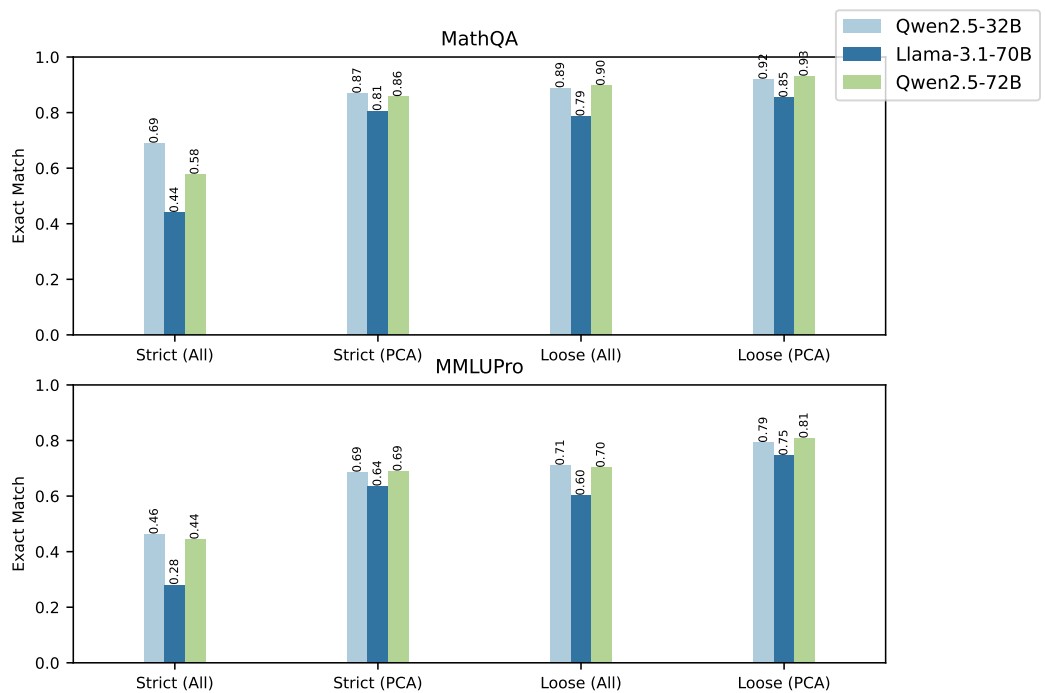

Figure 7: Large-Scale Models: Performance variation (strict and loose) of exact match scores for the Numeric Manipulation instruction category compared to its corresponding performance on *print_correct_answer* (PCA).

### A.7 INSTRUCTION SPECIFIC RESULTS

In this section, we report results at the individual instruction level across knowledge tasks.

### A.8 ERROR CLASSIFICATION

#### A.8.1 INFLUENCE OF PARAMETER SIZE

We report the performance on Full Benchmark for models from the Llama family and Qwen family of models (Figures 33 and 34). We observe a consistent pattern of improvements in instruction following-ability with increase in model capacity for the Llama family. However, this is not the case for Qwen family of models. Specifically, for some instructions like *print_correct_answer*, *print_correct_answer_label*, *sort_only_incorrect_answers* the Qwen 1.5B model outperforms 3B model. Qwen 3B model is better than Qwen 7B and 14B variants for the *print_correct_answer_append_string* instruction. We consistently see 32B and 72B variants outperforming other models by a significant margin.

### A.9 INFLUENCE OF DISTRACTORS

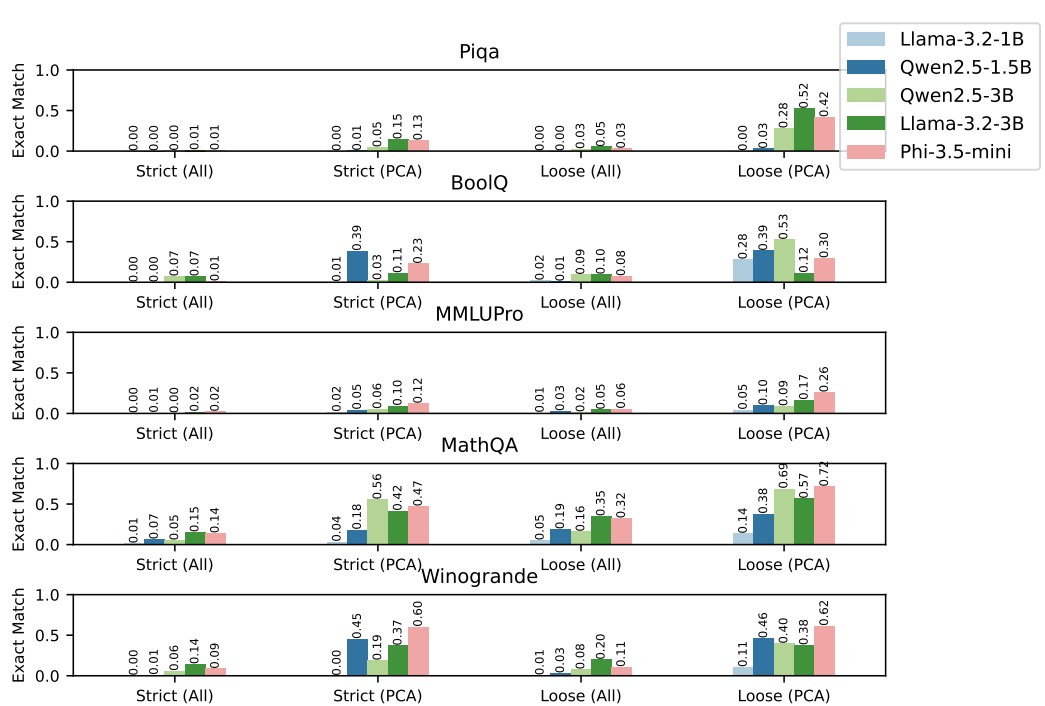

Figure 8: Small-Scale Models: Performance variation (strict and loose) of exact match scores for the String Manipulation instruction category compared to its corresponding performance on *print_correct_answer* (PCA).

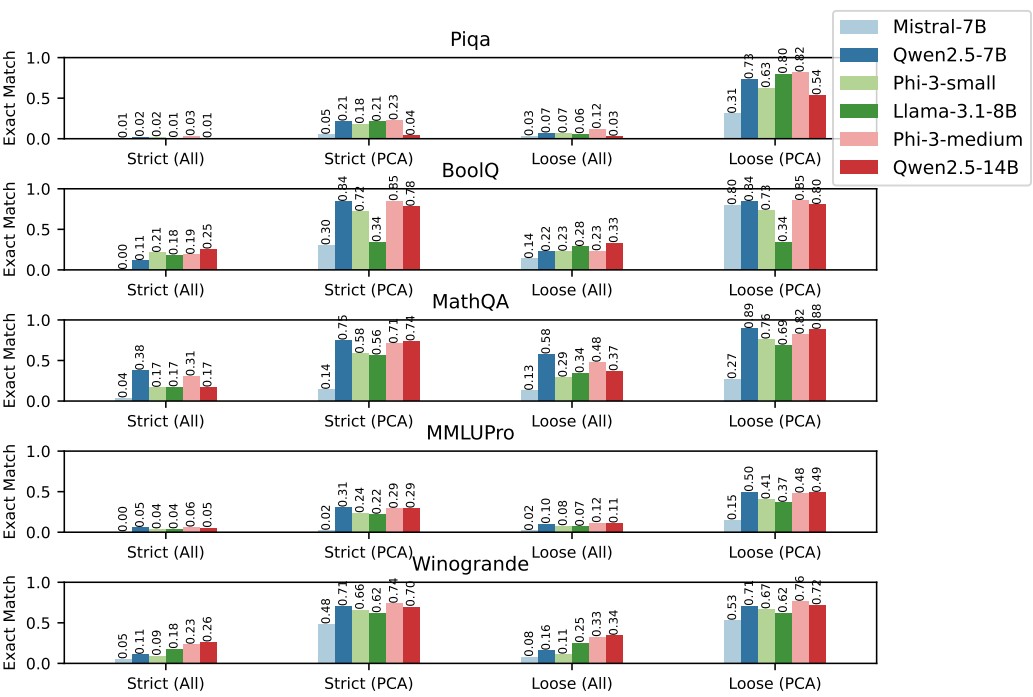

Figure 9: Medium-Scale Models: Performance variation (strict and loose) of exact match scores for the String Manipulation instruction category compared to its corresponding performance on *print_correct_answer* (PCA).

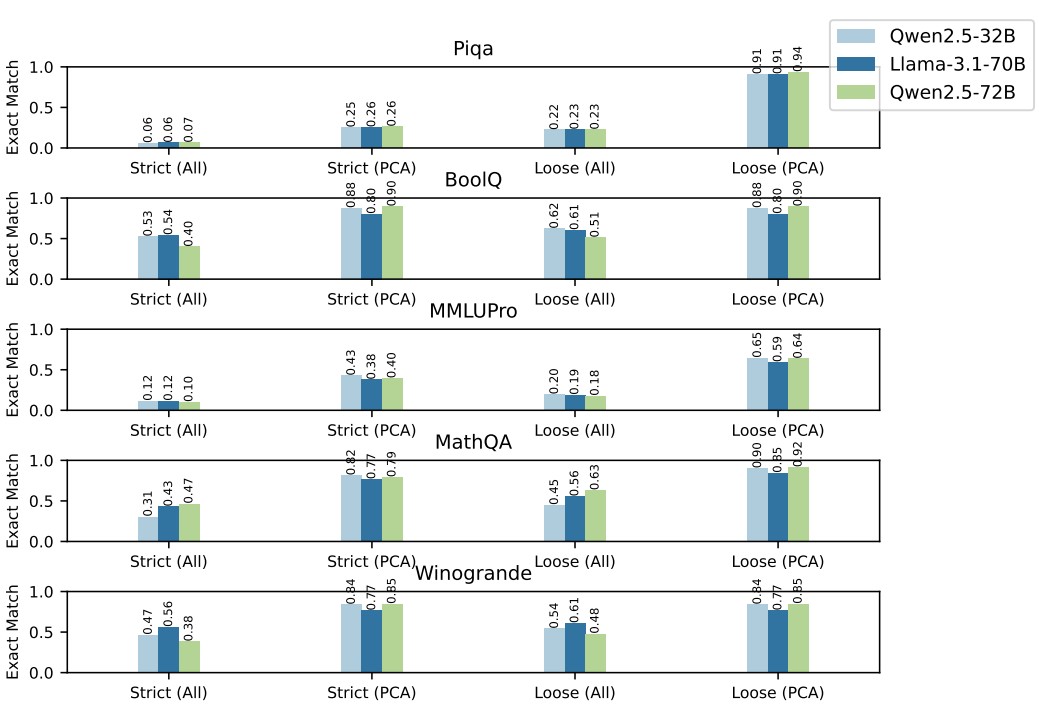

Figure 10: Large-Scale Models: Performance variation (strict and loose) of exact match scores for the String Manipulation instruction category compared to its corresponding performance on *print_correct_answer* (PCA).

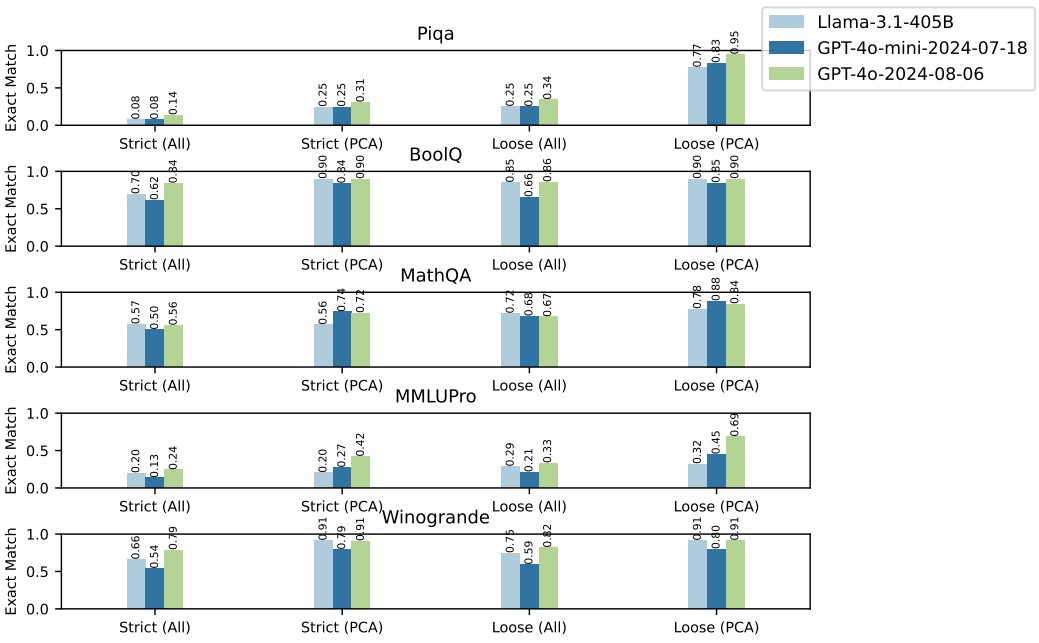

Figure 11: Frontier Models: Performance variation (strict and loose) of exact match scores for the String Manipulation instruction category compared to its corresponding performance on *print_correct_answer* (PCA).

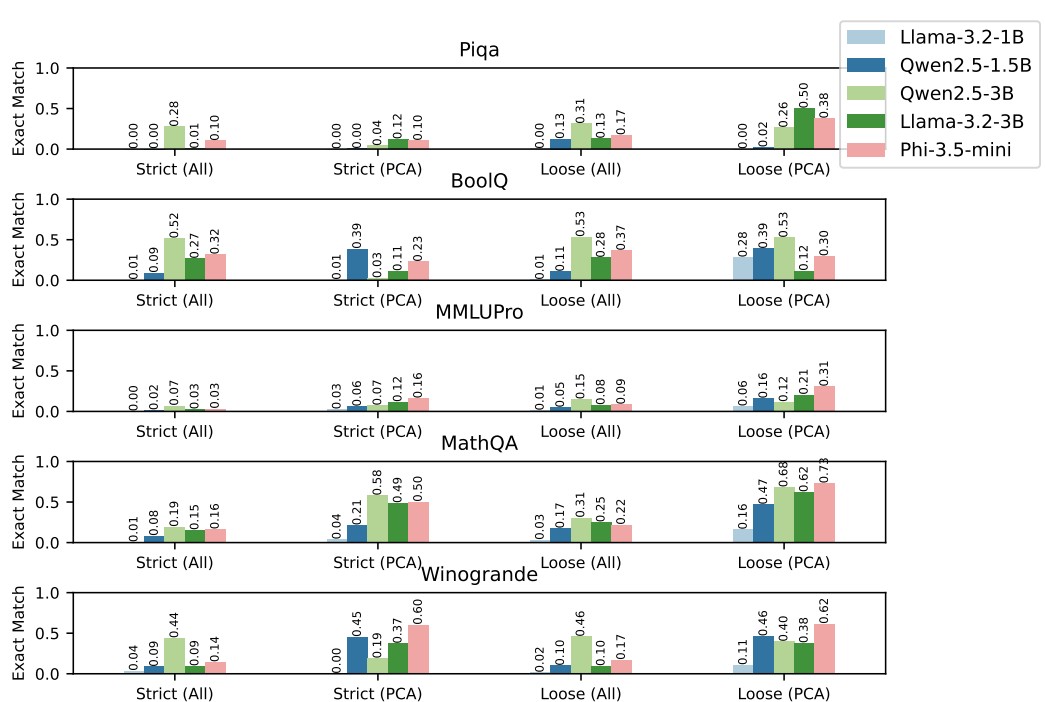

Figure 12: Small-Scale Models: Performance variation (strict and loose) of exact match scores for the Format Correct Answer instruction category compared to its corresponding performance on *print_correct_answer* (PCA).

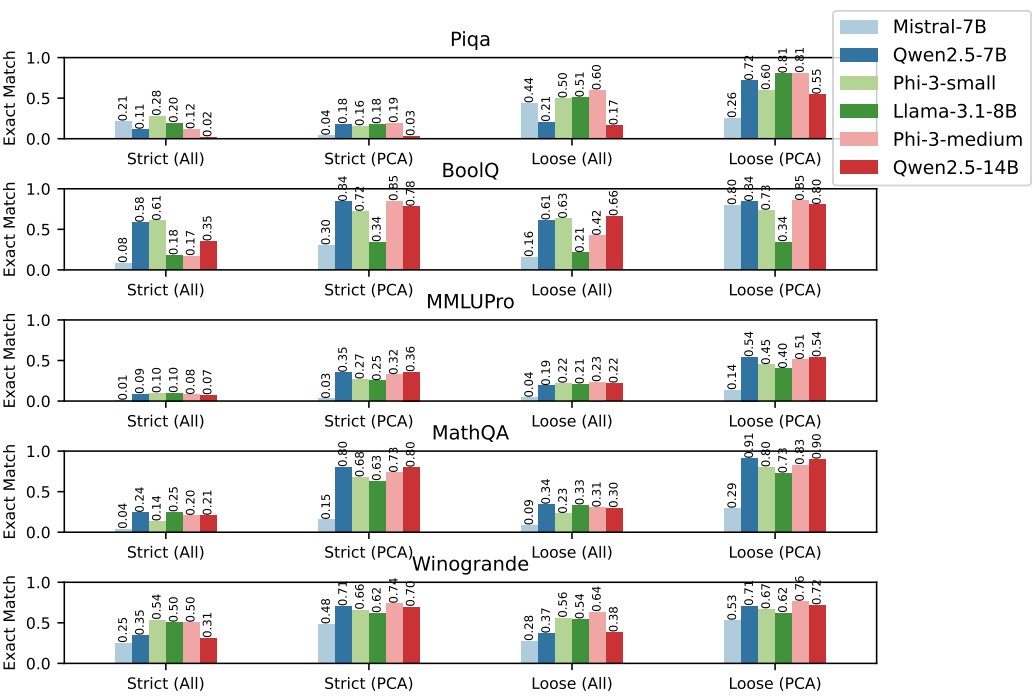

Figure 13: Medium-Scale Models: Performance variation (strict and loose) of exact match scores for the Format Correct Answer instruction category compared to its corresponding performance on *print_correct_answer* (PCA).

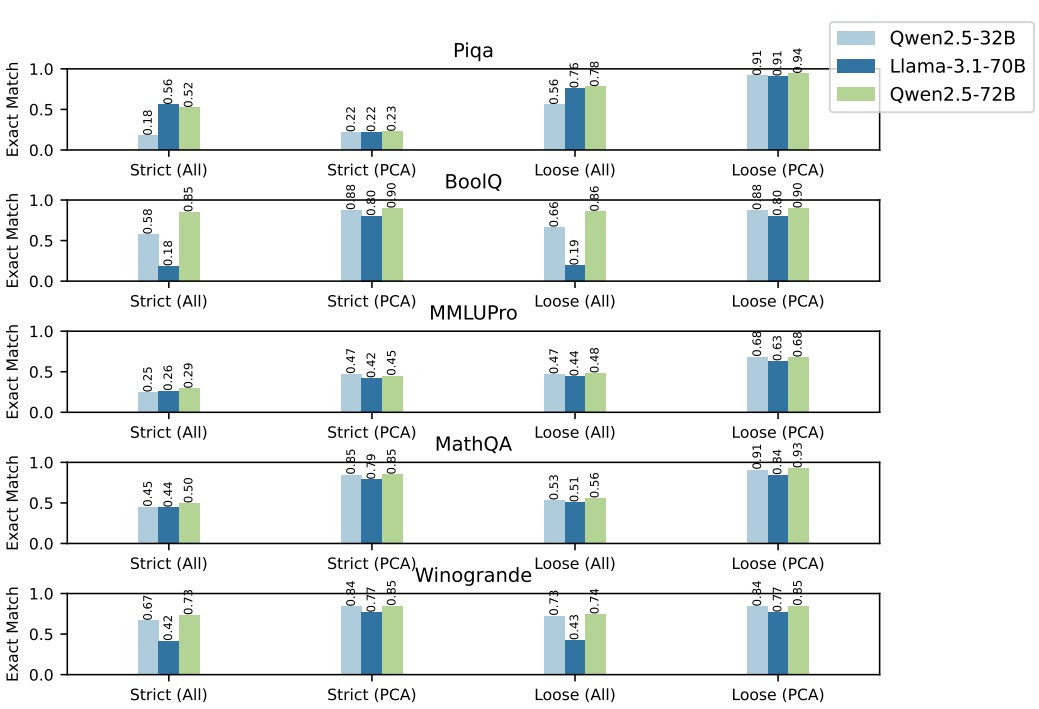

Figure 14: Large-Scale Models: Performance variation (strict and loose) of exact match scores for the Format Correct Answer instruction category compared to its corresponding performance on *print_correct_answer* (PCA).

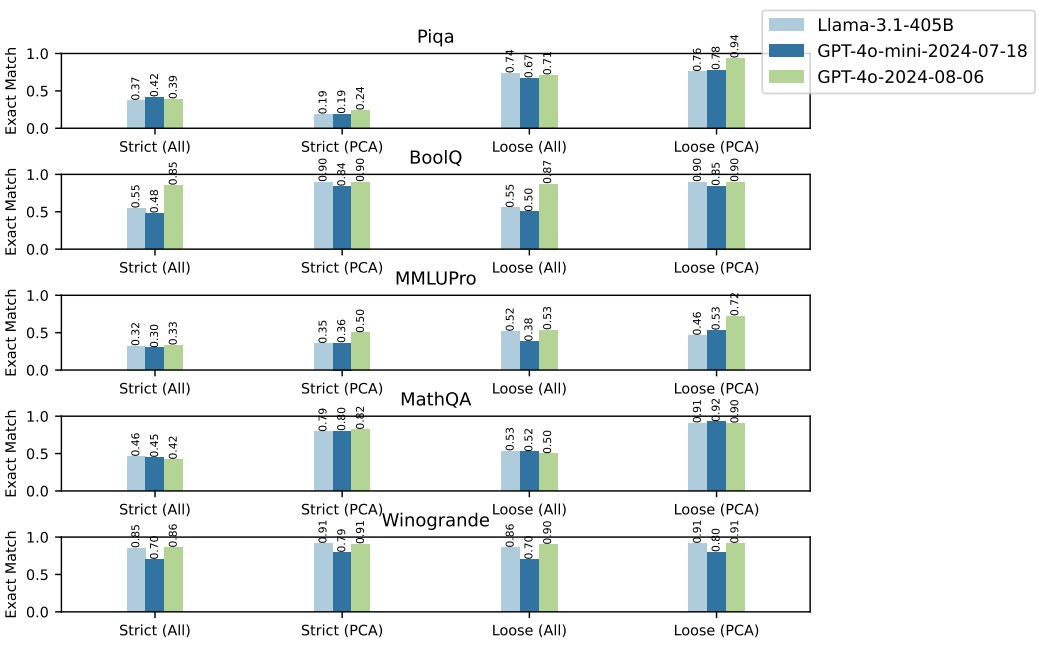

Figure 15: Frontier-Scale Models: Performance variation (strict and loose) of exact match scores for the Format Correct Answer instruction category compared to its corresponding performance on *print_correct_answer* (PCA).

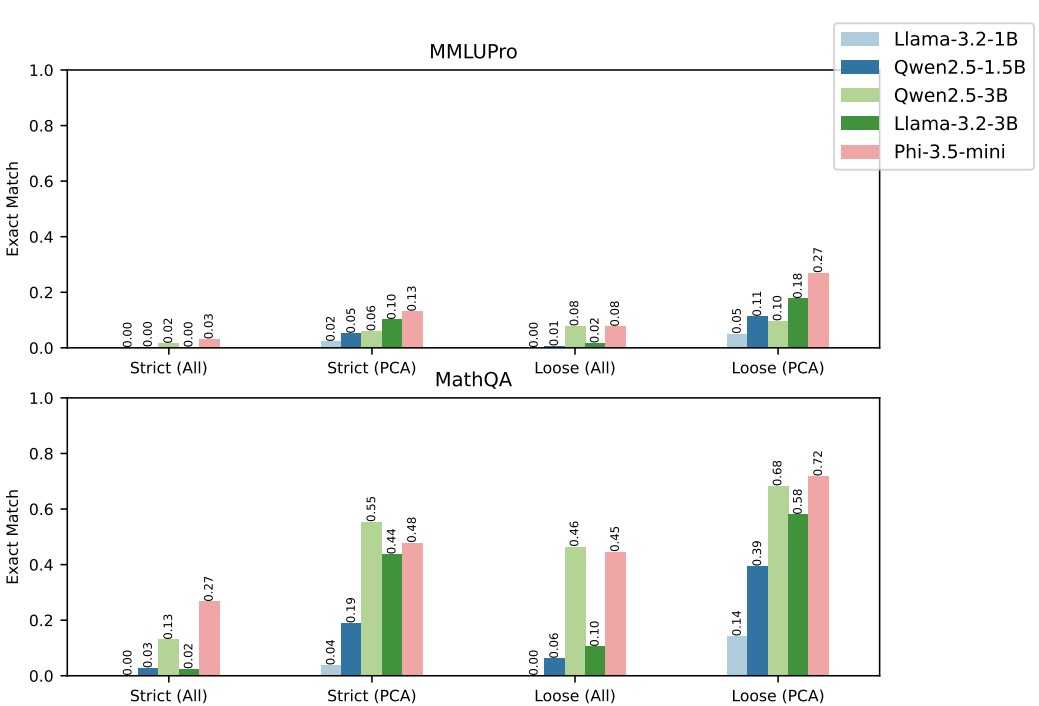

Figure 16: Small-Scale Models: Performance variation (strict and loose) of exact match scores for the Operations on List instruction category compared to its corresponding performance on $print\_correct\_answer$ (PCA).

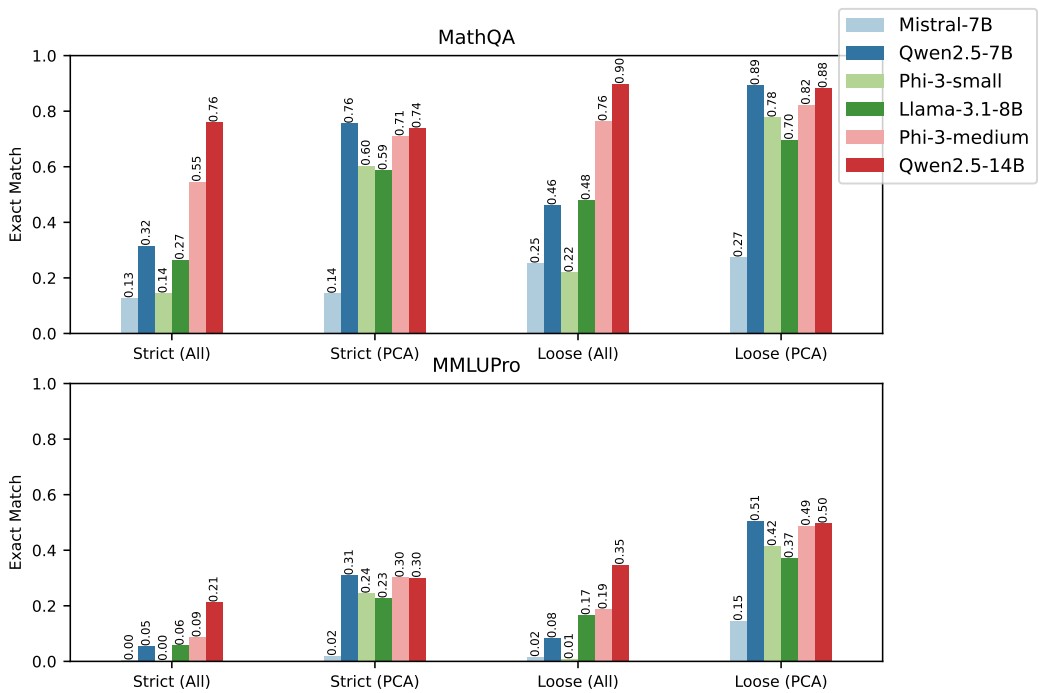

Figure 17: Medium-Scale Models: Performance variation (strict and loose) of exact match scores for the Operations on List instruction category compared to its corresponding performance on $print\_correct\_answer$ (PCA).

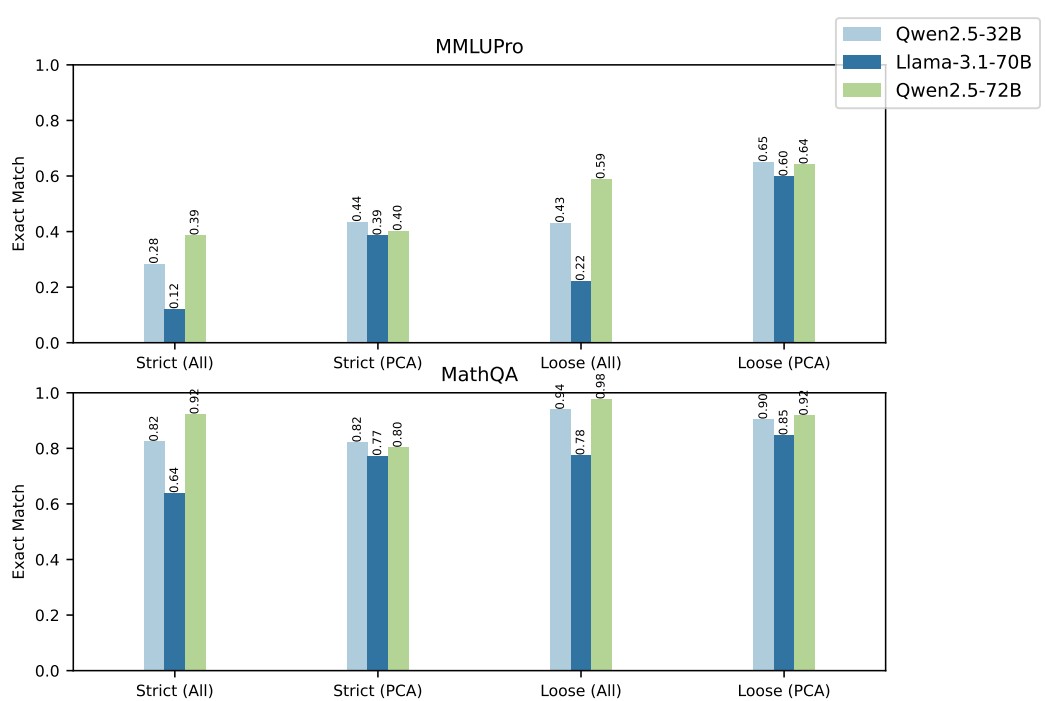

Figure 18: Large-Scale Models: Performance variation (strict and loose) of exact match scores for the Operations on List instruction category compared to its corresponding performance on *print_correct_answer* (PCA).

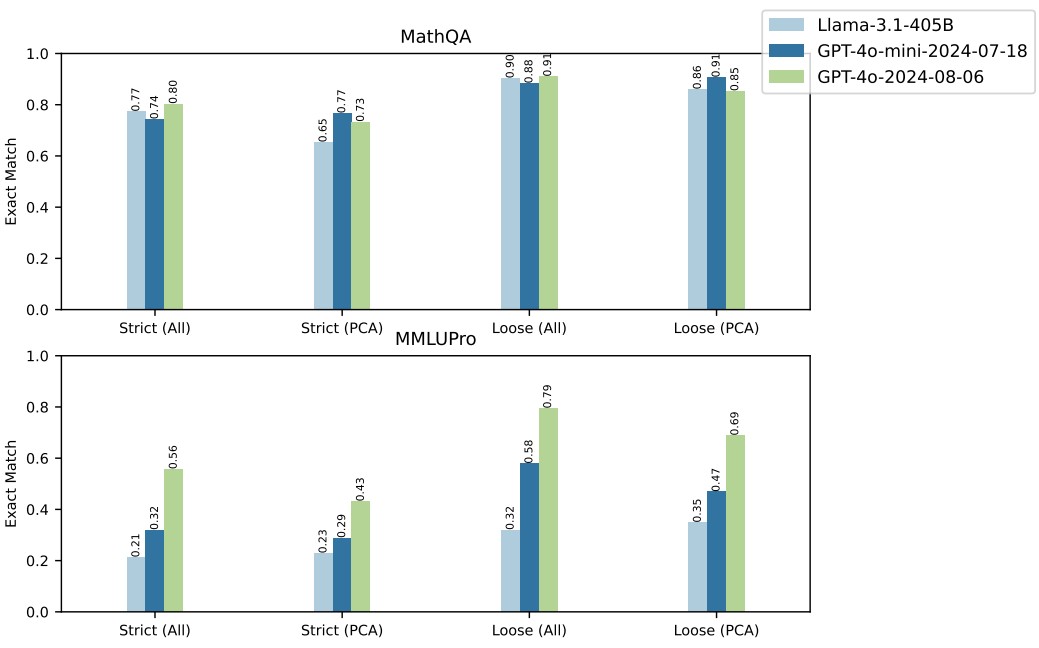

Figure 19: Frontier Models: Performance variation (strict and loose) of exact match scores for the Operations on List instruction category compared to its corresponding performance on *print_correct_answer* (PCA).

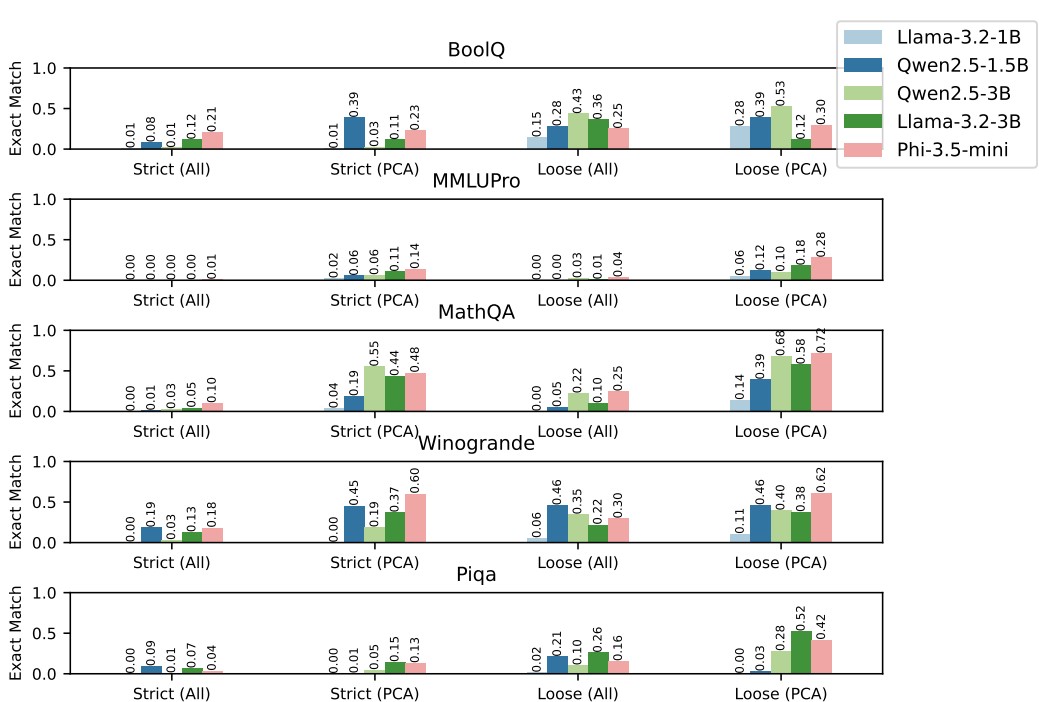

Figure 20: Small-Scale Models: Performance variation (strict and loose) of exact match scores for the Operations on List (Conditional) instruction category compared to its corresponding performance on $print\_correct\_answer$ (PCA).

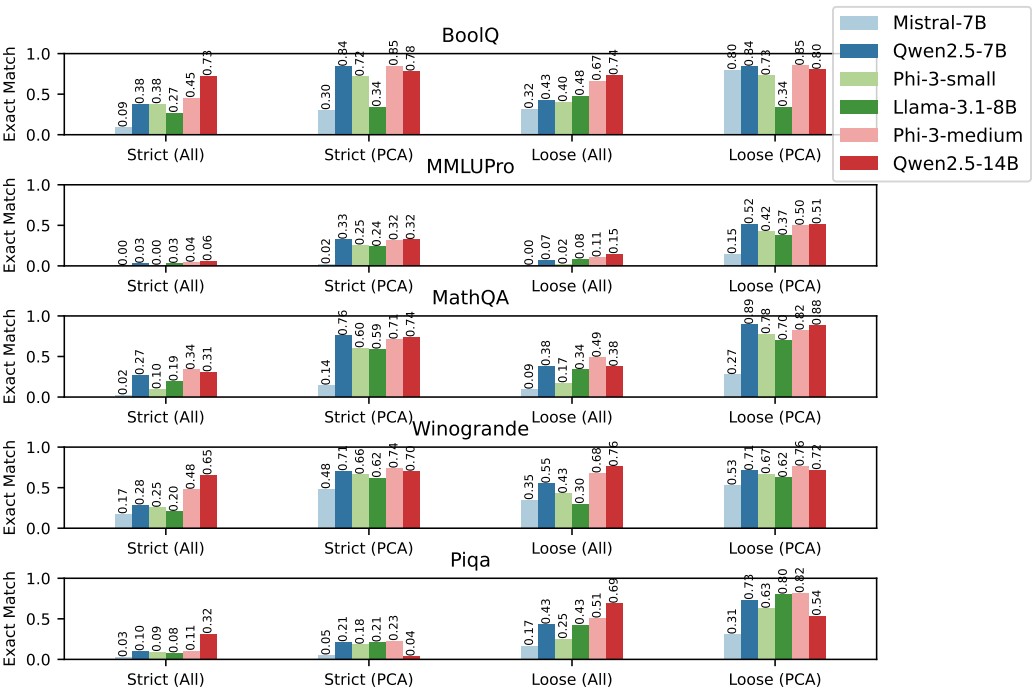

Figure 21: Medium-Scale Models: Performance variation (strict and loose) of exact match scores for the Operations on List (Conditional) instruction category compared to its corresponding performance on $print\_correct\_answer$ (PCA).

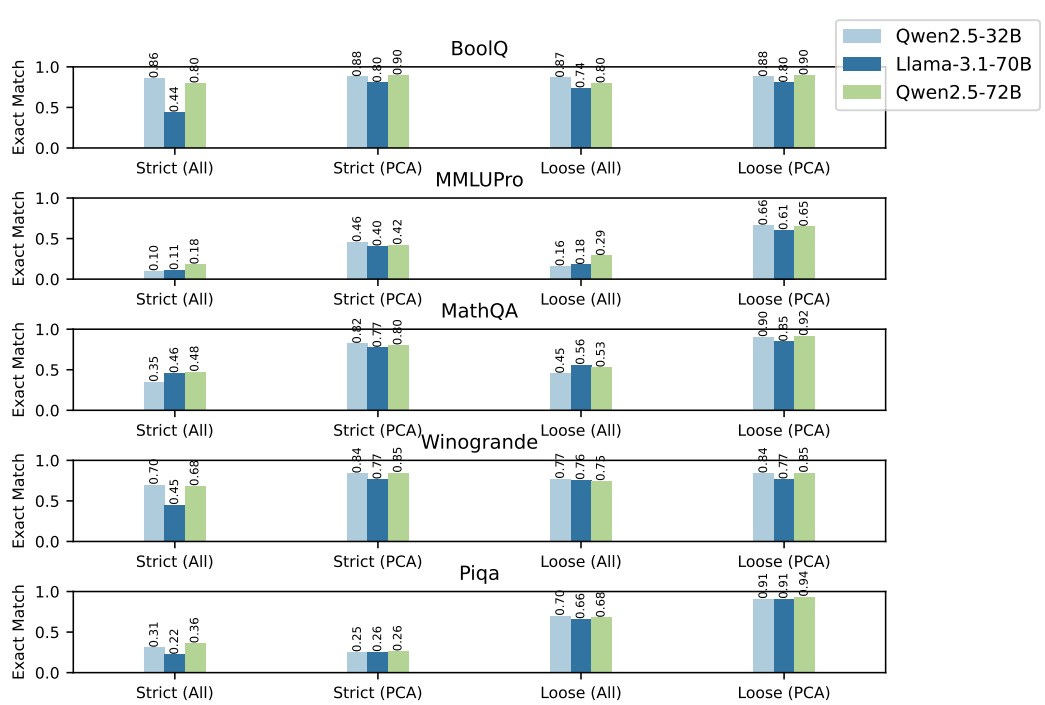

Figure 22: Large-Scale Models: Performance variation (strict and loose) of exact match scores for the Operations on List (Conditional) instruction category compared to its corresponding performance on $print\_correct\_answer$ (PCA).

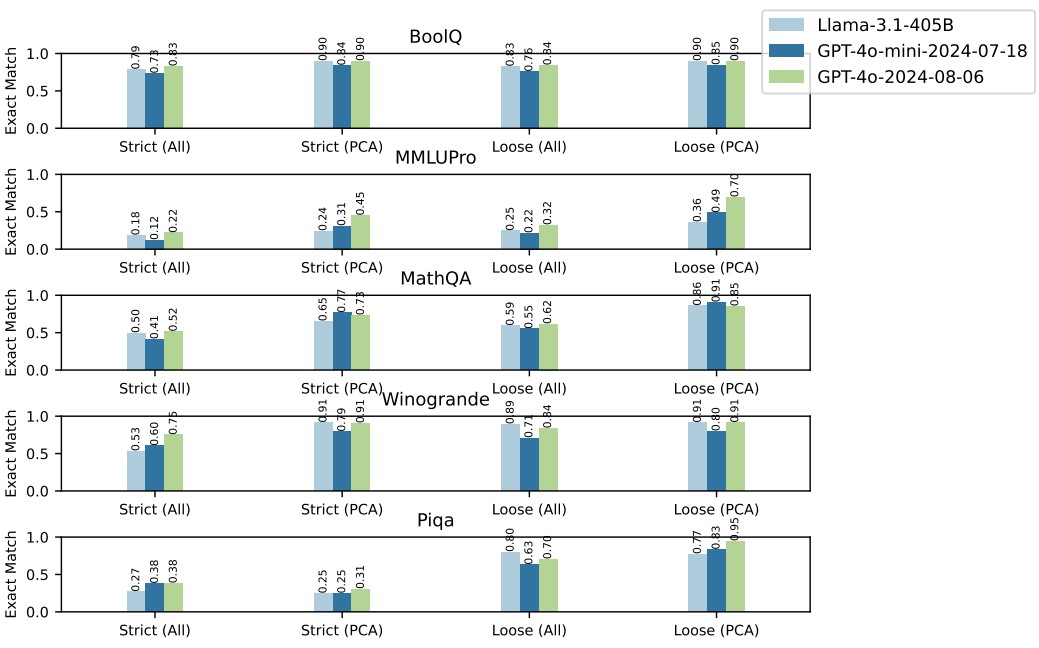

Figure 23: Frontier Models: Performance variation (strict and loose) of exact match scores for the Operations on List (Conditional) instruction category compared to its corresponding performance on $print\_correct\_answer$ (PCA).

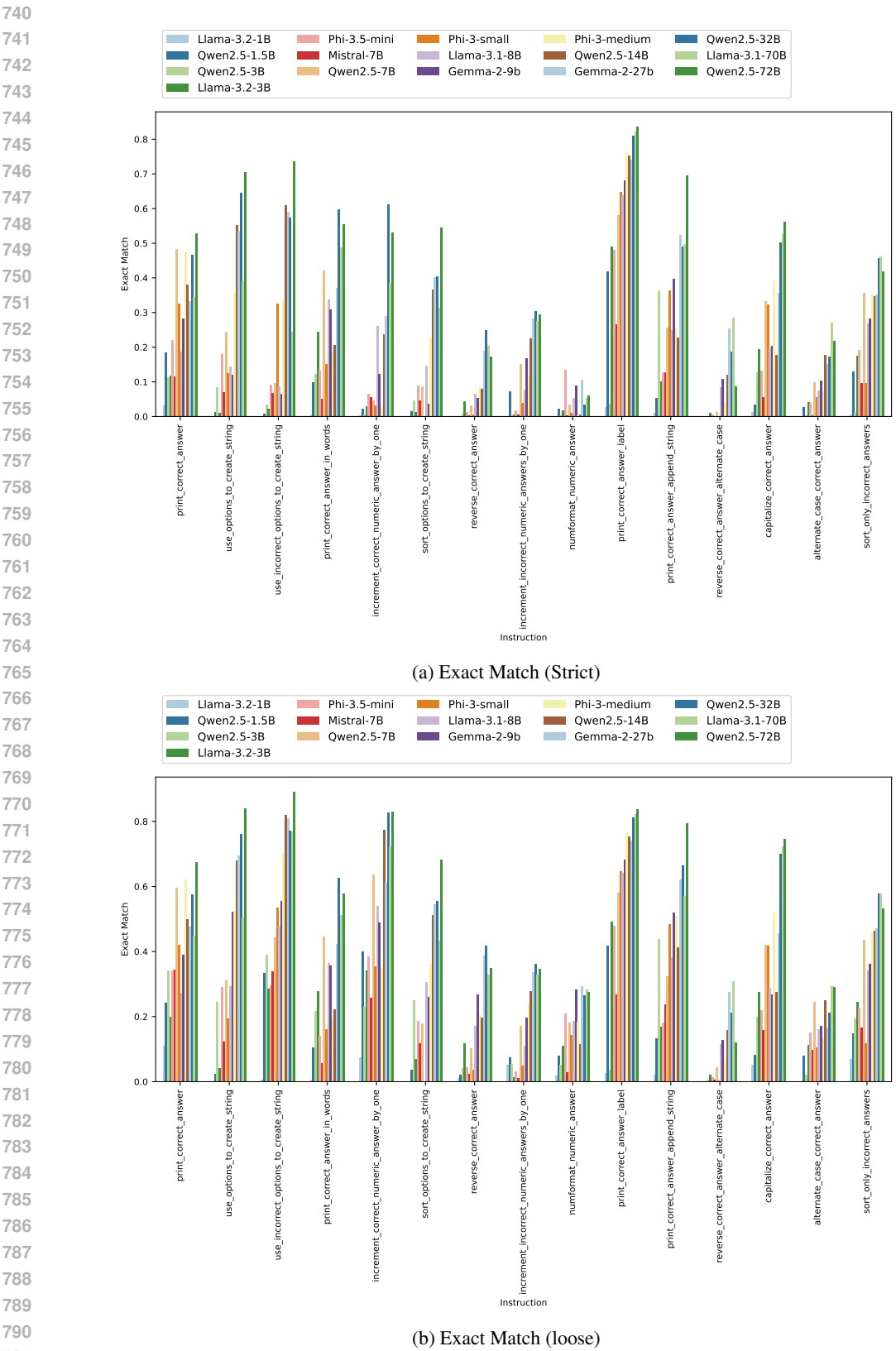

(a) Exact Match (Strict)

(b) Exact Match (loose)

Figure 24: Performance variation of exact match scores on individual instructions across models on Full Benchmark

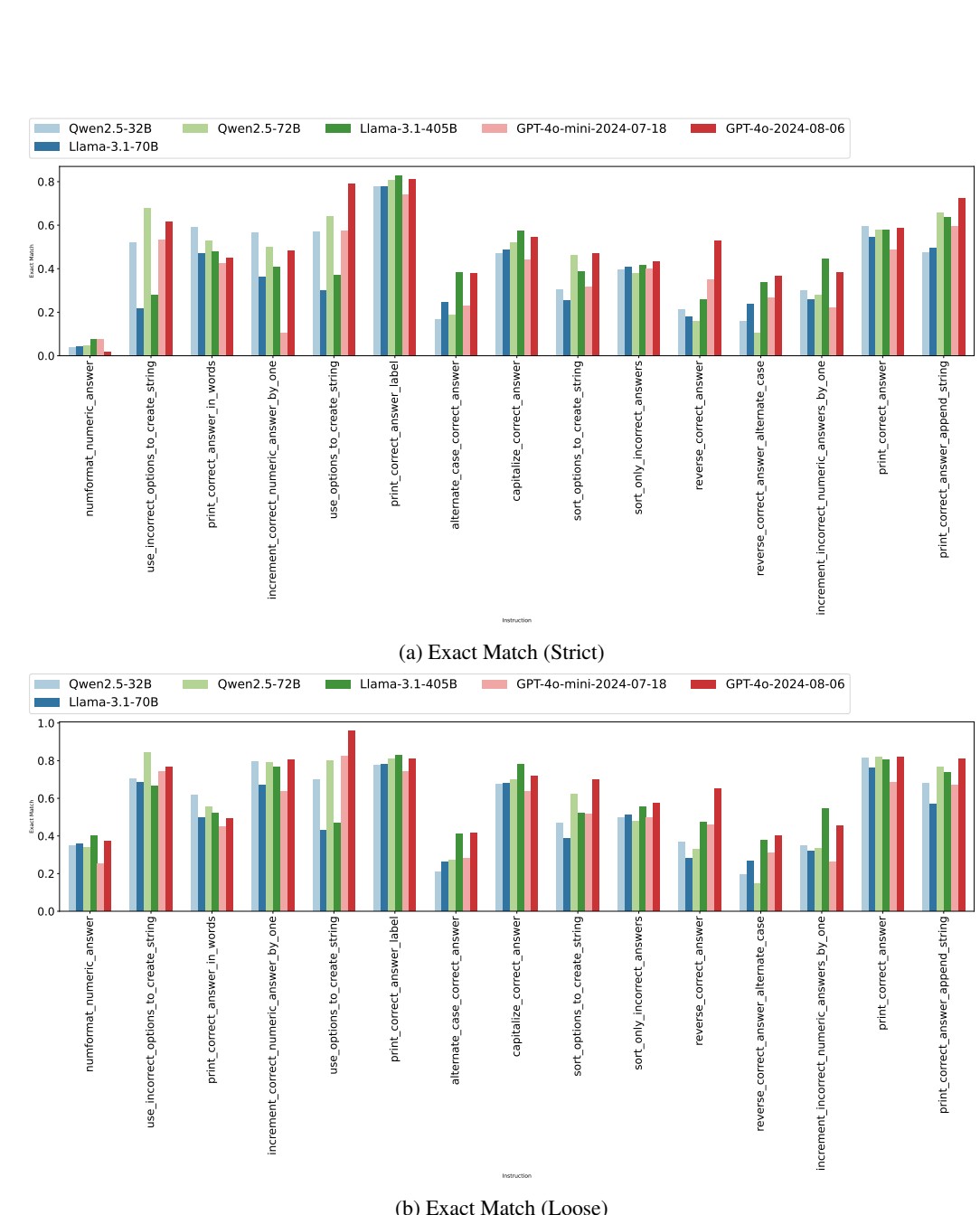

(a) Exact Match (Strict)

(b) Exact Match (Loose)

Figure 25: Performance variation of exact match scores on individual instructions across models on Lite Benchmark

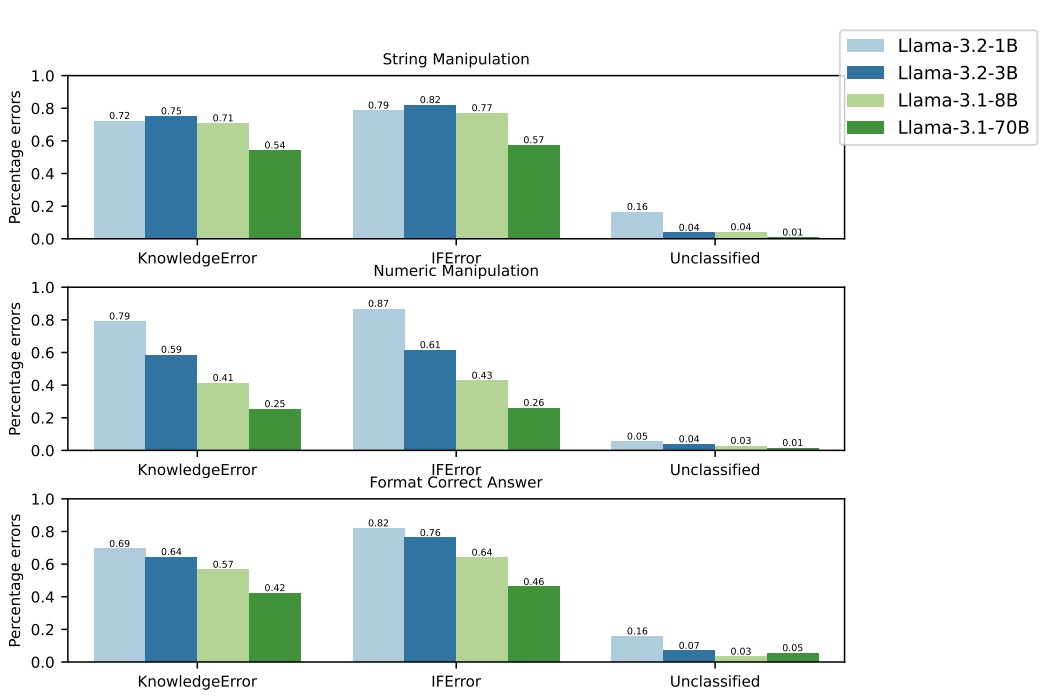

Figure 26: Llama model family: Knowledge Errors and IFErrors

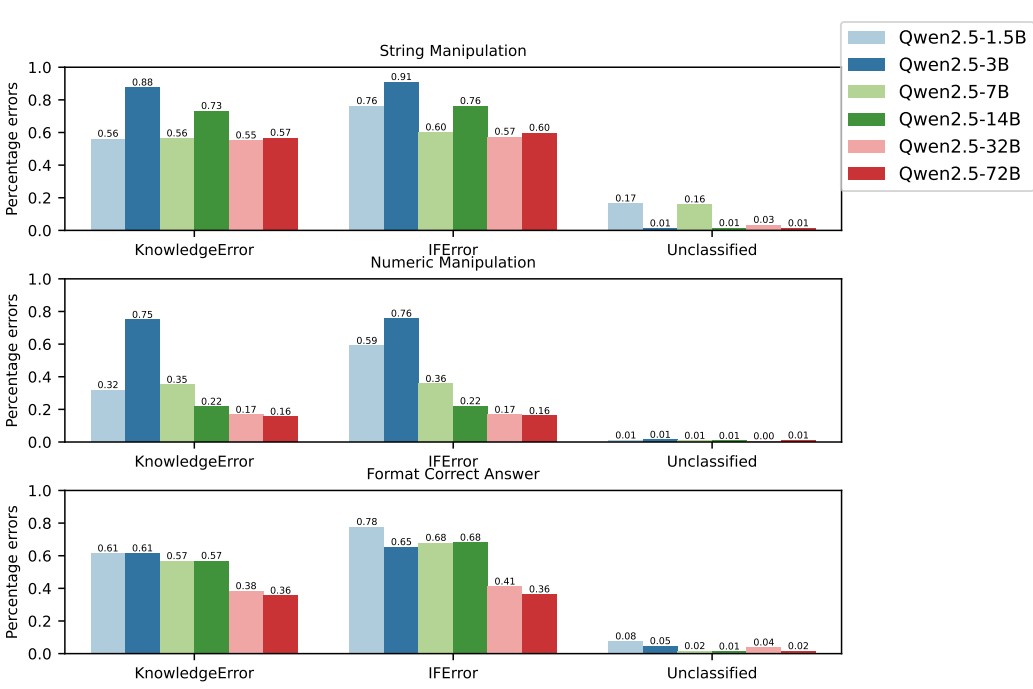

Figure 27: Qwen model family: Knowledge Errors and IFErrors

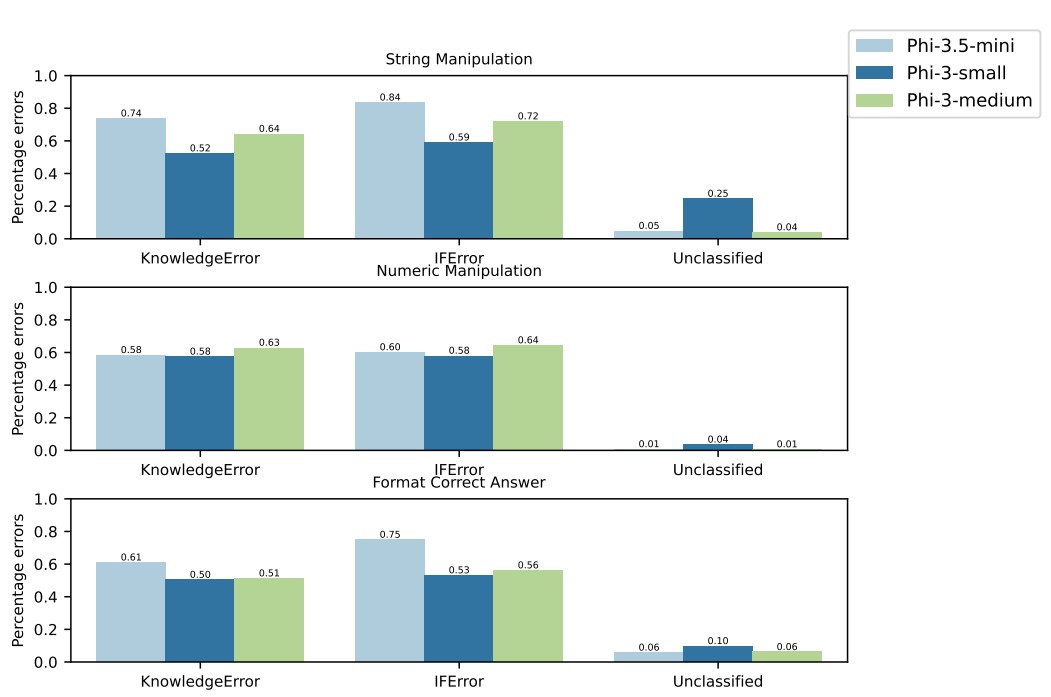

Figure 28: Phi model family: Knowledge Errors and IFErrors

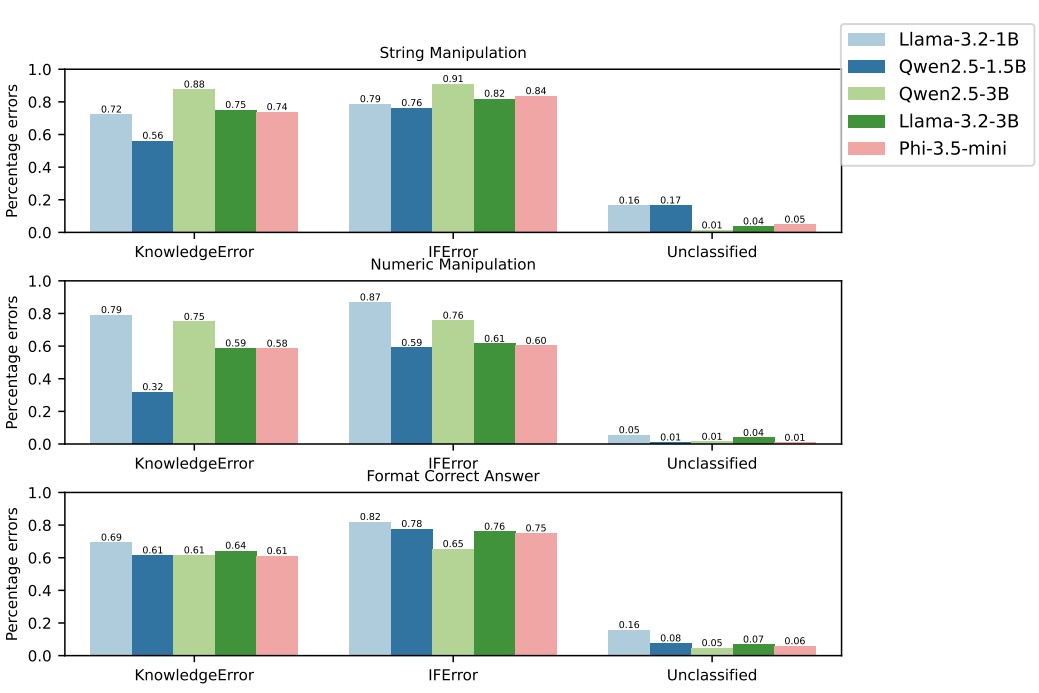

Figure 29: Small Scale Models: Knowledge Errors and IFErrors

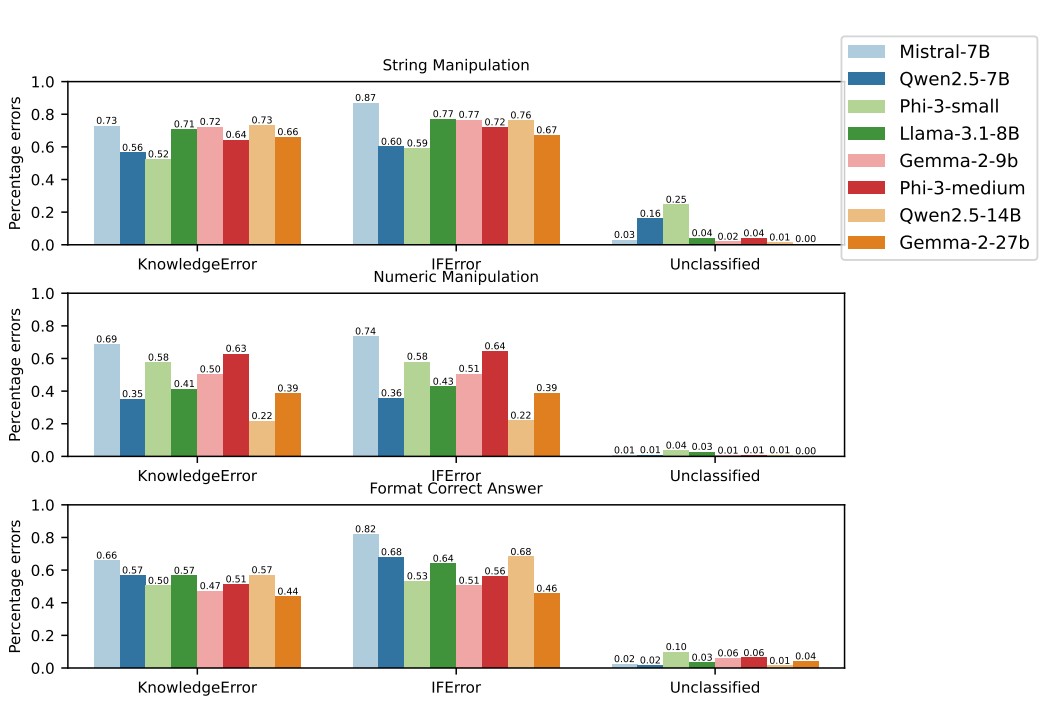

Figure 30: Medium Scale Models: Knowledge Errors and IFErrors

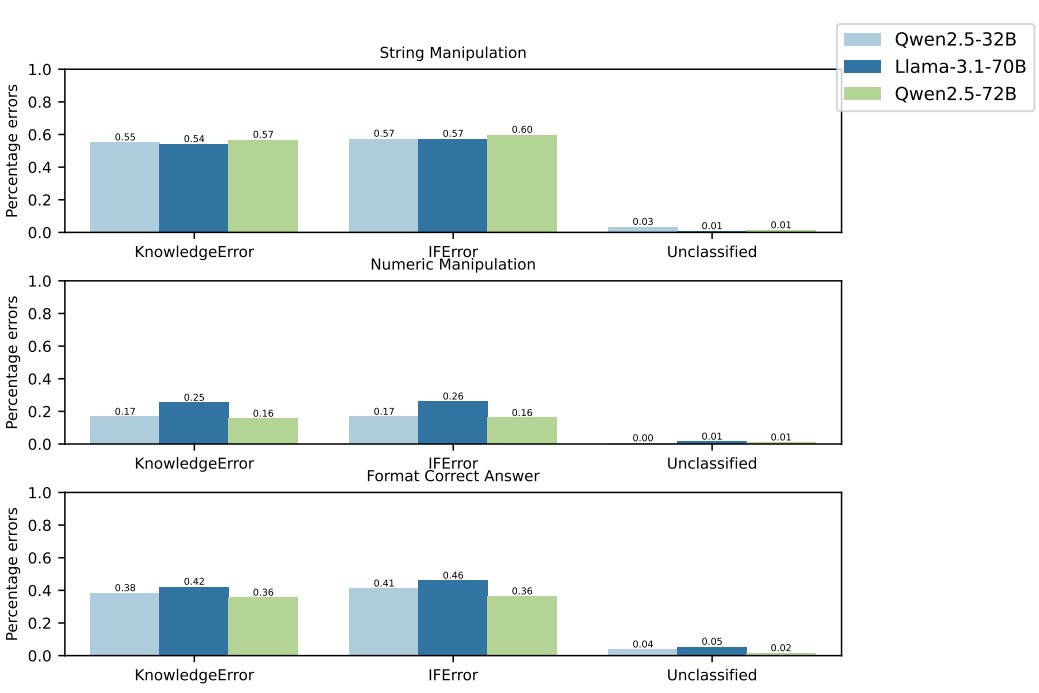

Figure 31: Large Scale Models: Knowledge Errors and IFErrors

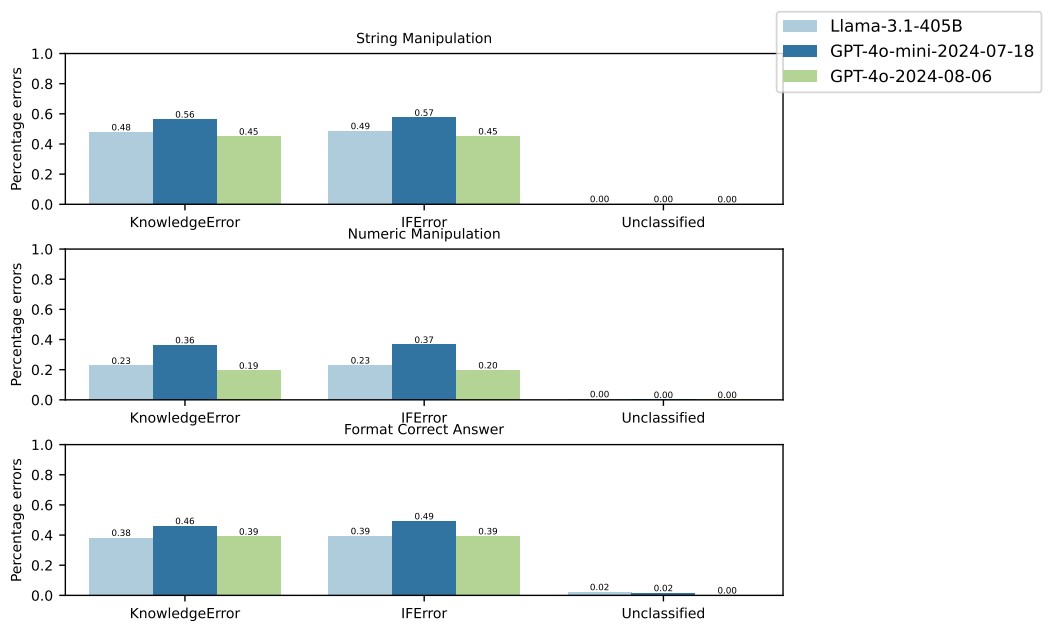

Figure 32: Frontier Models: Knowledge Errors and IFErrors

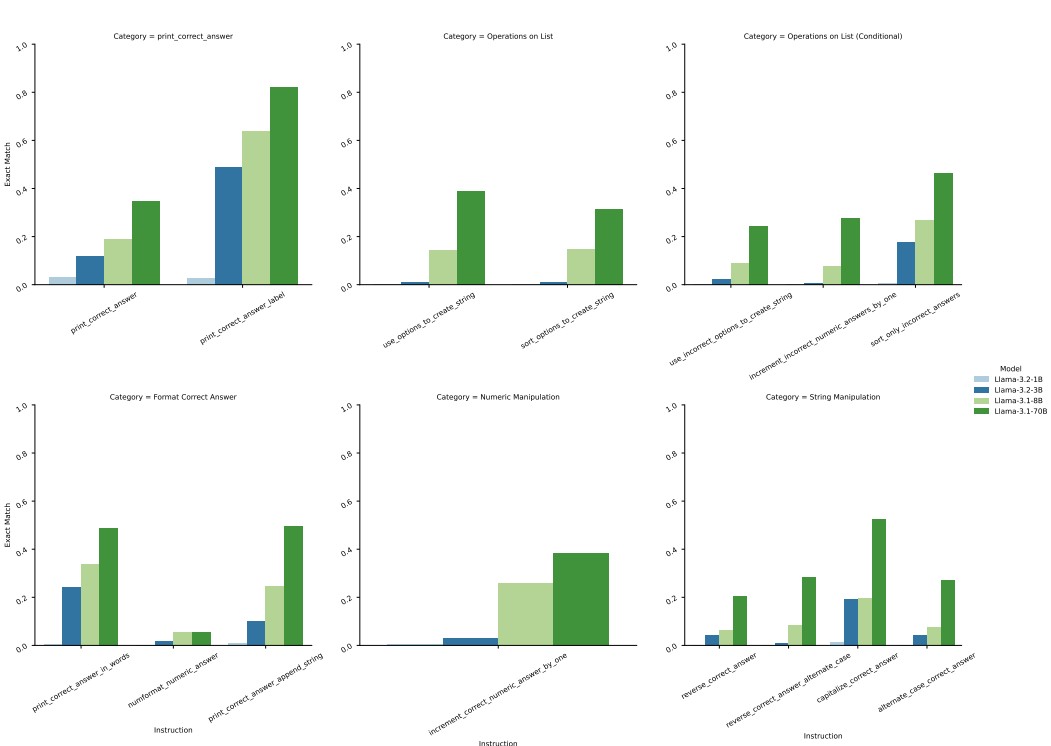

Figure 33: Performance variation (strict) of exact match scores for different instruction categories for Llama family of models

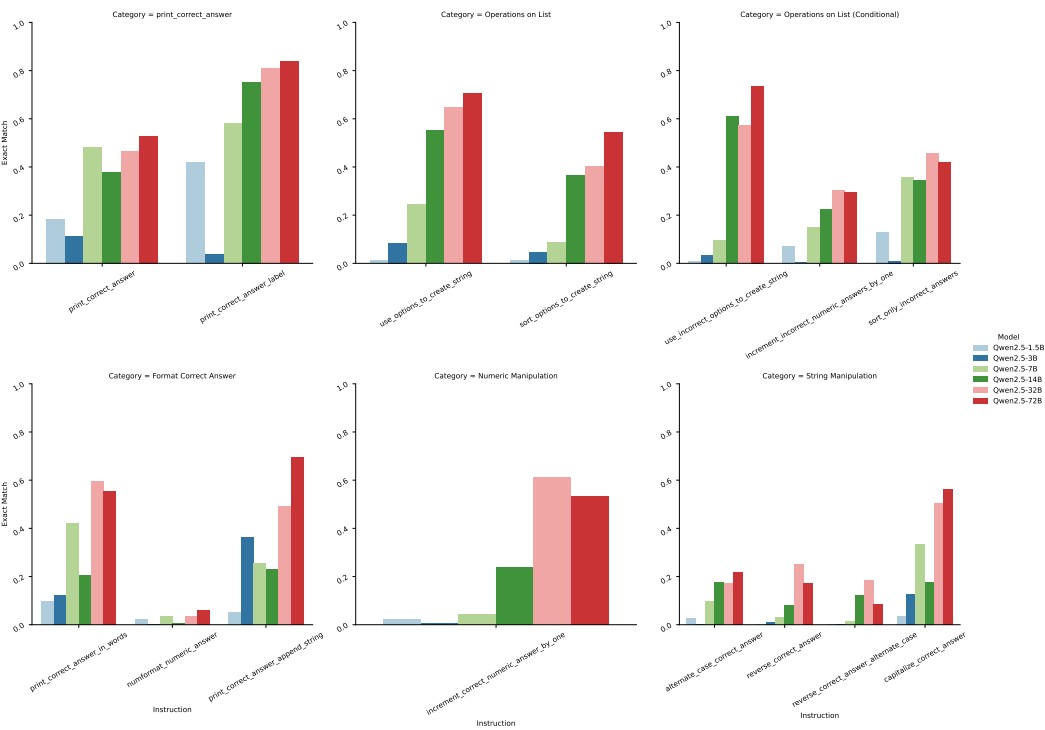

Figure 34: Performance variation (strict) of exact match scores for different instruction categories for Qwen family of models

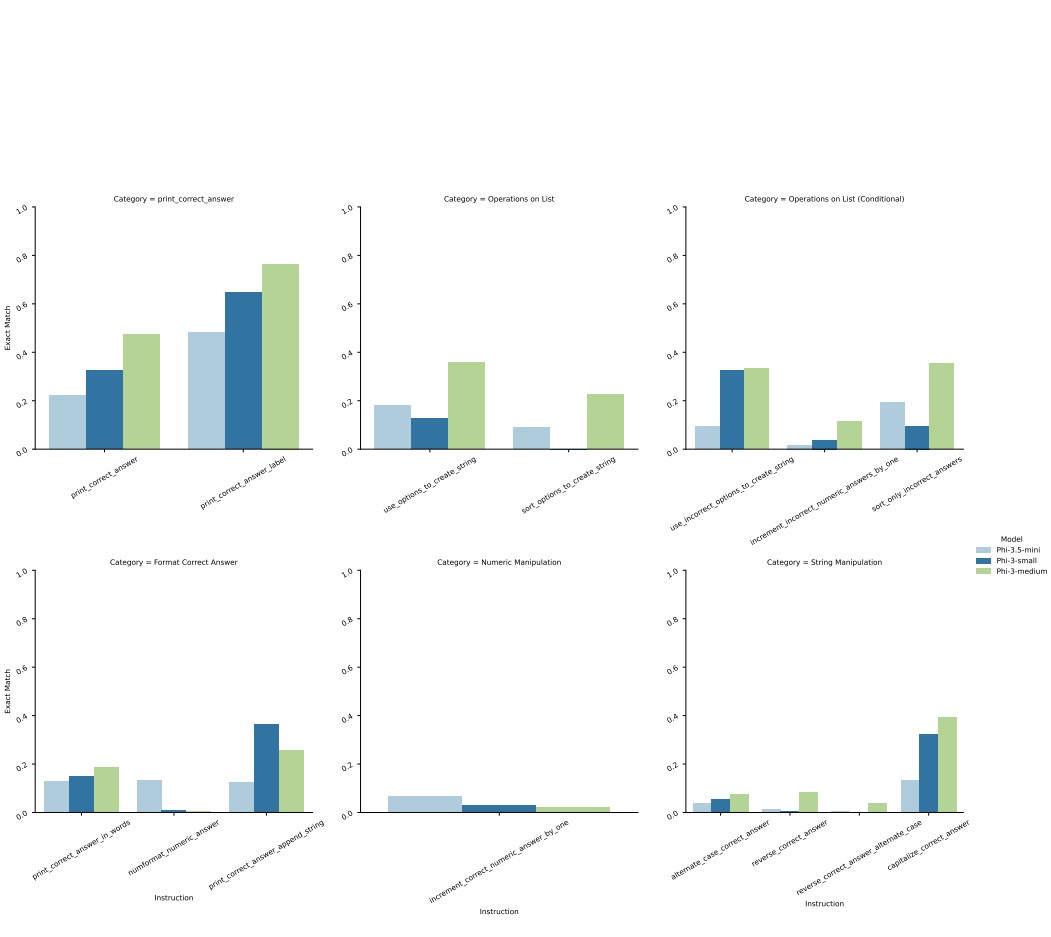

Figure 35: Performance variation (strict) of exact match scores for different instruction categories for Phi family of models

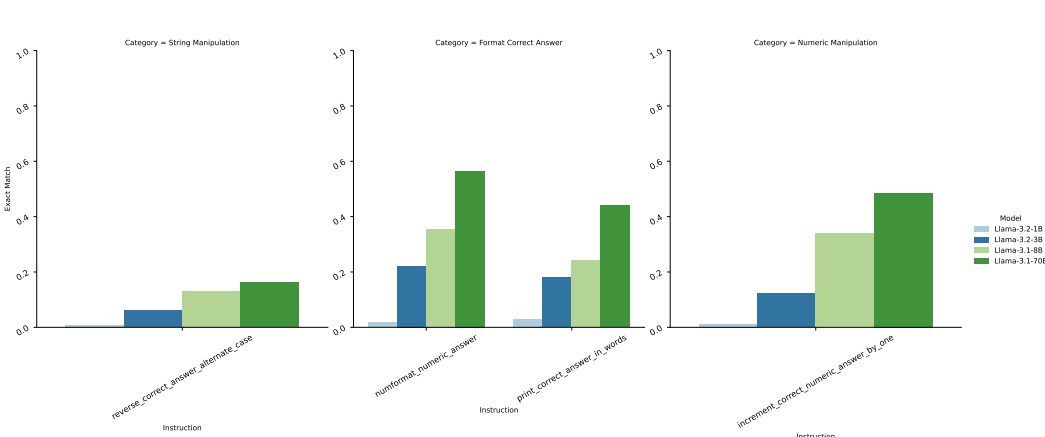

Figure 36: Performance variation (strict) of exact match scores for different instruction categories for Llama family of models on No Instruction following subset

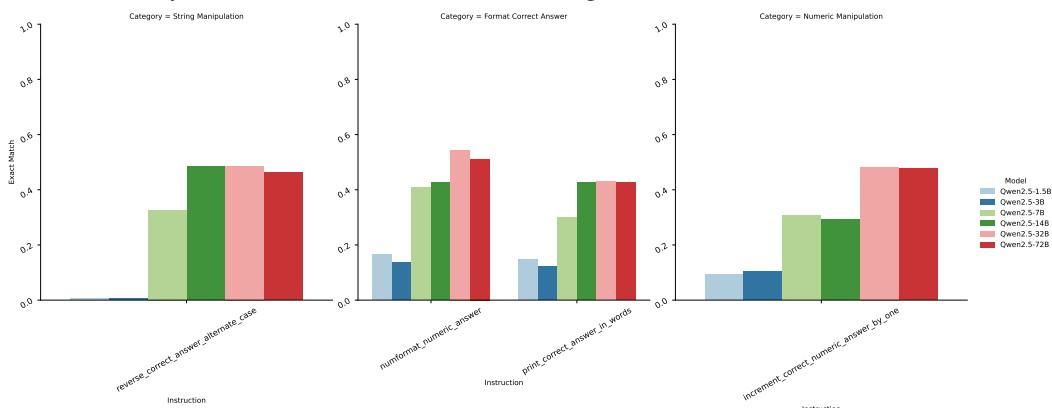

Figure 37: Performance variation (strict) of exact match scores for different instruction categories for Qwen family of models on No Instruction following subset

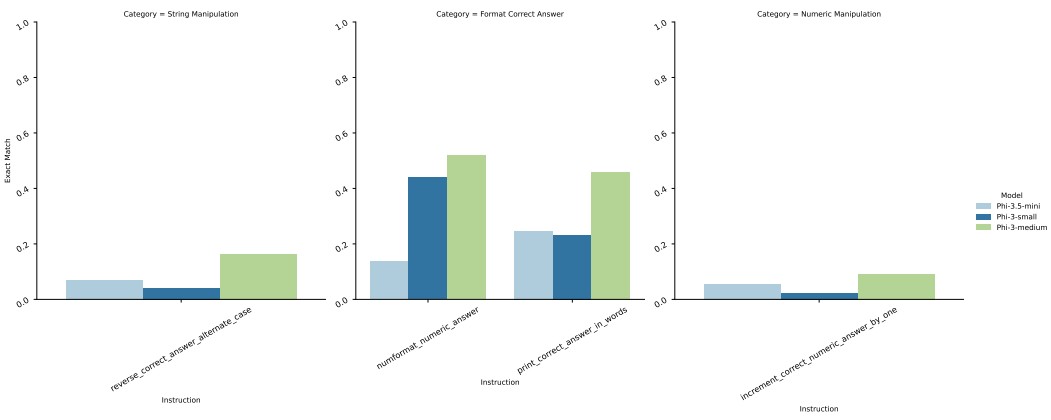

Figure 38: Performance variation (strict) of exact match scores for different instruction categories for Phi family of models on No Instruction following subset

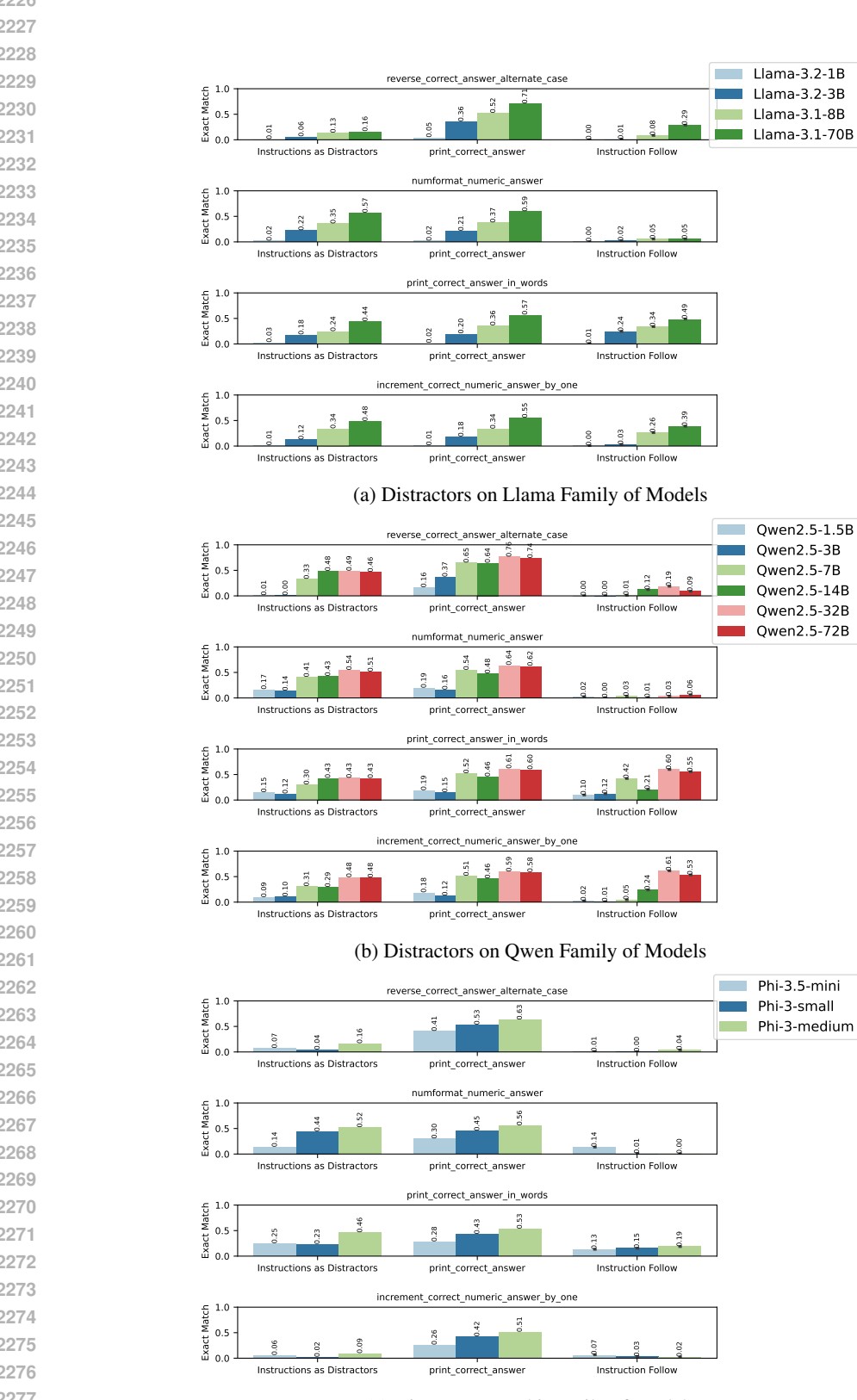

(a) Distractors on Llama Family of Models

(b) Distractors on Qwen Family of Models

(c) Distractors on Phi Family of Models

