# OpenReview forum: "Evaluating the Instruction-following Abilities of Language Models using Knowledge Tasks"
_ICLR.cc/2025/Conference — Submitted to ICLR 2025_

### Official Review · Reviewer_Z8cJ · 2024-10-31

**Soundness:** 1
**Presentation:** 2
**Contribution:** 1
**Rating:** 3
**Confidence:** 4

**Summary:**

The paper aims to evaluate the instruction-following abilities of language models within knowledge-based tasks by developing a benchmark that integrates instruction adherence and task performance. This approach attempts to measure both knowledge task success and instruction-following compliance using various publicly available large language models.

**Strengths:**

The methodology uses a range of instruction types and tasks, aiming to comprehensively assess model capabilities in both knowledge retrieval and instruction adherence.

**Weaknesses:**

1. **Limited Innovation:**
   The proposed benchmark shares considerable overlap with existing methods, particularly with IFEval, making it difficult to distinguish any unique contributions. Given that previous work has established benchmarks and instructions in similar contexts, this work falls short in terms of innovation.

2. **Unclear Figures:**
   Several figures, such as Figure 1 and Figure 4, are difficult to interpret due to unclear resolution or layout. Additionally, the figures on pages 8 and 9 might be better suited for the appendix, as they do not contribute substantively to the main discussion.

3. **Insufficient Ablation Studies:**
   The paper includes limited ablation experiments, which leaves the proposed benchmark's robustness and effectiveness underexplored. A deeper exploration of instruction categories and configurations would provide a stronger foundation for the benchmark's utility.

4. **Lack of Practical Implications:**
   The results from this benchmark do not provide any novel insights or actionable guidance for the training and development of future models. In its current state, the benchmark primarily reiterates known limitations of LLMs in instruction-following without advancing understanding in a way that could influence future model training approaches or benchmarks.

**Questions:**

Why did the authors choose to assess instruction-following and knowledge in a combined format? For instance, if instruction-following is the main focus, why not evaluate using IFEval first and then separately test knowledge tasks (such as MMLU) for a clearer analysis? This work combines them, which others have worked to disentangle, potentially obscuring instruction-following performance and knowledge retrieval.

---

> ### Author Response · Authors · 2024-11-23
> **Reviewer Question: Why study instruction-following with Knowledge?**
>
> Thank you for the question -- as we discuss in our related work, existing benchmarks either do not assess task performance when studying instruction following, or do so in ways that are not subjective assessments by LLMs (which have their own biases and limitations). In the case of IFEval, for example, if the task requires a model to answer the question in three words -- simply answer "the the the" would get a perfect score. We believe this is a major limitation of disentangling task performance from instruction-following as real-world performance is always a reflection of both. We use knowledge tasks as existing LLMs do perform well on these tasks (and give us a baseline performance for each model) -- adding simple instructions (instructions which are otherwise considered easy) leads to very observable drop in performance. Our goal was not to show that LLMs cannot follow, instructions but to provide a benchmark against which improvements can be objectively studied. For each model, we include the print correct answer text (PCA) and print correct answer label scores  -- contrasting instruction following scores against these disentangles knowledge and instruction-following performance. We would like to further state that we also plan to continuously extend this benchmark.

---

> ### Author Response · Authors · 2024-11-23
> **Practical Implications & Innovation**
>
> Our goal was to release a benchmark that could be used for studying instruction following with verifiable outputs. Constraints such as formatting style, length are harder to verify along with task performance (not just instruction performance) and there has been additional work in that space as you mentioned.
>
> However, work on assessing instruction-following with verifiable answers is limited. Our list of instructions demonstrate: 1. Simple changes, such as answering with text instead of labels results in a drop in performance at all model scales (and families). 2. Simple tasks of counting, concatenation, conditional exclusion/inclusion/application as well as, using distracting instructions all result in significant drop in performance.
>
> To the best of our knowledge, there is no prior work that demonstrates this with verifiable results and benchmarking of current models. We hope the reviewer agrees that this is valuable and non-verfiable instructions can lead to inflated assessments of instruction-following performance.  We further, agree that the list of instructions could be increased (our goal is continuously maintain and improve the collection through open-source contributions).

---

> ### Author Response · Authors · 2024-11-23
> **Additional Ablation Studies**
>
> Thank you for the feedback. We've now included additional sections that study model families -- effects of scale (within a family), and effects of distractors within a model family at different scales.
>
> **Performance within Model Families**: We report the performance on Full Benchmark for models from the Llama family and Qwen family of models in Figures in the updated appendix. We observe a consistent pattern of improvements in instruction following-ability with increase in model capacity for the Llama family. However, this is not the case for Qwen family of models. Specifically, for some instructions like *print_correct_answer, print_correct_answer_label, *sort_only_incorrect_answers the Qwen 1.5B model outperforms 3B model. Qwen 3B model is better than Qwen 7B and 14B variants for the print_correct_answer_append_string instruction. We consistently see 32B and 72B variants outperforming other models by a significant margin.
>
> **Distractors**:  We report details of how different model families (Llama, Qwen and Phi) are affected by distractors, at different scales. We find that the Llama family and Phi of models are extremely distracted by non-applicable instructions that require reversing and casing of text (eg: such instructions are inapplicable on numerical data), and report a drop of nearly 75-78% while Qwen family of models (at all scales) is relatively robust to such distractors. On the other hand, distractor instructions that are based on numeric operations lead to a minor drop in performance in Llama and Qwen models but still affect Phi family of models significantly.

---

> ### Author Response · Authors · 2024-11-23
> **Unclear Figures**
>
> We apologize for the this oversight and inconvenience caused. We are in the process of regenerating the figures for better rendering and we will further update the rebuttal version shortly.

---

> > ### Comment · Reviewer_Z8cJ · 2024-12-02
> >
> > I have reviewed your feedback and have decided to maintain my initial scores for now. Thank you for your input.

---

> > > ### Author Response · Authors · 2024-12-02
> > > **Acknowledgement**
> > >
> > > Thank you for your response - is there anything we can help address that would help improve the scores?
> > > We've included additional ablations/experiments as requested. We also hope the motivation of the work is clearer through our rebuttal. Please let us know if there's anything we can help clarify.

---

### Official Review · Reviewer_Aeeb · 2024-11-04

**Soundness:** 3
**Presentation:** 1
**Contribution:** 2
**Rating:** 3
**Confidence:** 4

**Summary:**

This paper presents a benchmark for evaluating the instruction-following abilities of LLMs on knowledge tasks with automated error classification. The authors augment existing dataset knowledge datasets (e.g., MMLU-Pro, MathQA) with manually-crafted instructions to test the instruction following abilities of LLMs. Extensive evaluation across various LLMs demonstrate failures in instruction-following, even among frontier LLMs.

**Strengths:**

- The authors provide extensive evaluations of various models ranging in size on their proposed instruction-following benchmark.

- By augmenting existing knowledge datasets with manually constructed instructions, the authors can automatically classify errors as either knowledge errors or instruction-following errors.

**Weaknesses:**

- As a whole, the main paper includes a lot of implementation details, which is not necessarily a bad thing, but in this case takes up valuable space that would be better used to provide additional discussion and analysis of the experimental results, which are largely underdeveloped. For instance, sections 4.1 and 4.2 could be briefly summarized in the main paper with the bulk of the details moved to the appendix.

- The automatic error classification is rather underdeveloped in the main paper and lacks sufficient analysis. For instance, only a few results in the main paper (Figure 1 and Table 4) include IC and KTS scores, and even then the subsequent discussion focuses primarily on variants of exact match. A lot of relevant analysis (e.g., on which instruction categories do models make the most instruction-following errors on average?) is hard to follow, in part because the layout of figure 2 makes it hard to compare across instruction categories.

- Similarly, the inclusion of new instruction classes (e.g., distractors, instructions that are conditional on the answer) is also rather surface-level. For instance, the conclusion drawn from section 4.3.3 on distractors is that they worsen model performance. More analysis is needed here (e.g., which models are the most susceptible to distractors, what kinds of distractors are most effective).

- The instructions categories are rather limited in comparison to existing work. Several of the instructions (e.g., alternate_case_correct_answer) play into known tokenization issues with LLMs [1]. Existing work such as IFEval include more practical instruction categories (e.g., length constraints, JSON formatting) [2].

- The writing quality needs to be improved. For instance, in the definition for “alternate_case_correct_answer”, “correction candidate answer” should be “correct candidate answer”. In section 3.5 line 268, the reader is referred to “Table 2 in the appendix” though table 2 is in the main paper. Likewise, some terminology is used inconsistently (e.g., print_correct_answer_text vs. print_correct_answer). This didn't affect readability too significantly so I did not factor this point into my decision assessment.

**Questions:**

- How can you measure correctness (in terms of knowledge) for instructions such as use_options_to_create_strings, where the output is seemingly independent of the correct answer?

- Could you verify that table 2 is correct? For instance, the definition of increment_incorrect_numeric_answers_by_one suggests that all answers should be indicated, not just the incorrect ones. Likewise, why is use_incorrect_options_to_create_string in the non-conditional operations on lists category when it seems to be inherently conditional on the correct answer?

- Related to figure 1, could you provide some insight into Qwen-2.5-1.5B’s performance? Why is the strict (PCA) score so similar between Qwen-2.5-1.5B and Qwen-2.5-3B, yet Qwen-2.5-1.5B’s strict (PCA label) score is significantly higher than Qwen-2.5-3B’s? Why are the knowledge and instruction-following error scores significantly higher for Qwen-2.5-1.5B than Qwen-2.5-3B even though the strict (PCA) scores are so similar?

- On average, what percent of errors can be automatically classified as either knowledge errors or instruction-following errors? Figure 1 leads me to believe that lots of errors escape categorization (e.g., Llama-3.2-1B achieves a strict (PCA) score of 2%, yet only 17% and 9% of instances can be classified as knowledge errors and instruction-following errors, respectively. Is this due to the high-precision nature of these metrics? If so, would supplementing with an LLM-based method for error categorization be better?

- Are certain categories of distractors (e.g., among the 5 instruction categories) more effective than others? Are some models (or family of models) more susceptible to certain categories of distractors than others?

[1] https://arxiv.org/abs/2410.19730

[2] https://arxiv.org/abs/2311.07911

I am willing to raise my score if my questions/concerns are adequately addressed.

---

> ### Author Response · Authors · 2024-11-23
> **Acknowledgement**
>
> Thank you very much for the detailed reading of our paper and your insightful comments. We hope to get back by addressing all your questions/comments shortly. In the meantime, we address some of your concerns in our responses below.

---

> > ### Author Response · Authors · 2024-11-23
> > **Distractor Analysis**
> >
> > We've now included additional sections that study model families -- effects of scale (within a family), and effects of distractors within a model family at different scales.
> >
> > **Performance within Model Families**: We report the performance on Full Benchmark for models from the Llama family and Qwen family of models in Figures in the updated appendix. We observe a consistent pattern of improvements in instruction following-ability with increase in model capacity for the Llama family. However, this is not the case for Qwen family of models. Specifically, for some instructions like *print_correct_answer, print_correct_answer_label, *sort_only_incorrect_answers the Qwen 1.5B model outperforms 3B model. Qwen 3B model is better than Qwen 7B and 14B variants for the print_correct_answer_append_string instruction. We consistently see 32B and 72B variants outperforming other models by a significant margin.
> >
> > **Distractors**: We report details of how different model families (Llama, Qwen and Phi) are affected by distractors, at different scales. We find that the Llama family and Phi of models are extremely distracted by non-applicable instructions that require reversing and casing of text (eg: such instructions are inapplicable on numerical data), and report a drop of nearly 75-78% while Qwen family of models (at all scales) is relatively robust to such distractors. On the other hand, distractor instructions that are based on numeric operations lead to a minor drop in performance in Llama and Qwen models but still affect Phi family of models significantly.

---

> ### Author Response · Authors · 2024-11-23
> **Limited Instruction Categories**
>
> **Comment on Instruction Categories** Our goal was to release a benchmark that could be used for studying instruction following with verifiable outputs. Constraints such as formatting style, length are harder to verify along with task performance (not just instruction performance) and there has been additional work in that space. However, assessing instruction-following with verifiable answers is limited. Our list of instructions demonstrate: 1. Simple changes of return text instead of labels results in a drop. 2. Simple tasks of counting, concatenation, conditional exclusion/inclusion/application as well as distracting instructions all result in significant drop in performance.
>
> To the best of our knowledge, there is no prior work that demonstrates this with verifiable results and benchmarking of current models. We hope the reviewer agrees that this is valuable and non-verifiable instructions can lead to inflated assessments of instruction-following performance.  We further, agree that the list of instructions could be increased (our goal is continuously maintain and improve the collection through open-source contributions).
>
> **Limitations of IFEval** We would like to further add, the instructions that we include, while simple, also contribute to "practical instructions" - for instance, IFEval includes instructions to check for capitalization -- however, it does not assess whether the target string being capitalized is correct or not. Using our benchmark, we demonstrate, that simple capitalization results in a 20% absolute drop in performance (as compared no capitalization) on models such as Llama 70B Instruct but the Qwen family of models aren't as affected by such an instruction. Such insights cannot be drawn using existing benchmarks. Another example is that, all models fail to correctly append strings to correct answers of the original knowledge task. IFEval instructions that require the inclusion of a prefix strings for Titles etc are similar in spirit but fail to account for task performance/correctness.

---

> ### Author Response · Authors · 2024-11-23
> **use_options_to_create_string**
>
> The instruction is as follows: *Create a string by concatenating the last character of every option value (not option label). If the last character is a special character (such as period, comma, quotation, etc) use the previous character.*
>
> Options values are shown to LLM in a fixed sequence and the results of this operation are thus deterministic and verifiable. We generate ground-truth data by implementing these actions in code.

---

> ### Author Response · Authors · 2024-11-23
> **Categorization error in Table 2 (use_incorrect_options_to_create_string)**
>
> We apologize for the oversight -- the instruction category for use_incorrect_options_to_create_string is indeed conditional and the main category label was misaligned for this row in LaTeX.

---

> ### Author Response · Authors · 2024-11-23
> **increment_incorrect_numeric_answers_by_one**
>
> Yes that is correct -- this instruction requires returning a list. It is conditional on the correct answer in the sense, that the list includes all options (incremented by one) *except* for the correct answer.

---

> ### Author Response · Authors · 2024-11-23
> **Typographical errors (Last comment in weakness)**
>
> We apologize of the typographical errors and we have incorporated changes and also proof-read the paper again. We shall be updating the rebuttal version of the PDF shortly.

---

> ### Author Response · Authors · 2024-11-27
> **Error analysis and Qwen performance**
>
> Thank you for pointing out the discrepancies with Figure 1 and the performance of Qwen.
> We have updated the rebuttal version with corrections and added details regarding the error analysis, and have also uploaded more figures showing errors across different model families. Here is the summary for your reference.
>
> **Qwen performance**
>
> 1. We took a closer look at the raw output from Qwen-2.5-3B and noted a consistent pattern of *Response: $Label* for the *print_correct_answer_label* (i.e.) extra $ symbol. While this could arguably be considered an instruction following error, we decided to modify our evaluation to not penalize this minor issue.
> With the new PCA label score, we see that Qwen-2.5-1.5B and Qwen-2.5-3B have comparable performance for PCA and PCA label.
>
> 2. We also realized that we had accidentally reported the strict version of the errors instead of the loose (that we had mentioned in the paper). This means in the updated version, our error detection sets continue to be high-precision, however the categorization of errors into the different buckets is high recall. Not doing so, would automatically result in the unclassified categorization becoming high recall, which is not helpful for error analysis.
>
> 3. We have modified Figure 1 (and Figure 5) to additionally show the unclassified score, as well as the updated error analysis scores. We find that the Knowledge and IF-errors of Qwen-2.5-1.5B and Qwen-2.5-3B are comparable (within 4-5% of each other) , leading us to believe that the Qwen-2.5-3B model does not offer superior instruction following capabilities compared to its 1.5B counterpart.
>
>
> **Error Analysis**
>
> Additionally, from the updated Figure 1, we observe that the errors decrease as the inherent model capability and size increases. Note that incorrect answers could correspond to both knowledge and instruction following errors. The figure also shows that we capture most errors.
> We observe larger models making fewer errors ($20\%-80\%$ reduction) for the Llama and Qwen models. The Phi model family however does not show this trend, calling for a closer look at their instruction training methodology.
> We also perform a deep-dive on the error distribution for each instruction category across model scales. We observe that models make the most errors on string manipulation tasks, and model scale does little to mitigate this. For the other categories, errors reduce as the model size increases.
> All these figures are available in the appendix of the revised pdf.

---

### Official Review · Reviewer_q5jM · 2024-11-07

**Soundness:** 2
**Presentation:** 2
**Contribution:** 2
**Rating:** 3
**Confidence:** 4

**Summary:**

This paper proposes to evaluate the instruction-following abilities of LLMs using Knowledge Tasks. It adopts several popular benchmarks and add rules on top of them to evaluate instruction following capability and content correctness at the same. It includes details of how to design these instructions and evaluation results of popular models including GPT-4, etc.

**Strengths:**

* This paper proposes a benchmark that both evaluate instruction following capability and content verification, which does not draw too much attention in the community before.
* This paper includes details regarding how to design the instructions and benchmarks, and run a lots of experiments to benchmark  different models in various scales.

**Weaknesses:**

* This paper ignores one key related work [1]. [1] instructs the model to verbalize the task label with words aligning with model priors to different extents, adopting verbalizers from highly aligned (e.g., outputting “positive” for positive sentiment), to minimally aligned (e.g., outputting “negative” for positive sentiment). During evaluations, [1] uses task labels after manipulations, which in fact evaluates both the instruction following capability and conduct content verification. In theory, this technique can be used for any classification task. However, such a related work was not discussed at all in this paper, indicating that Table 1 in this paper does not provide a comprehensive reviews to the community.


* Although authors run lots of experiments, it is hard to draw insights from their plots. For example, it is more interesting to plot model performance in the same family (e.g. LLaMA 3.1 in different sizes) to see how scaling can help solve this task instead of plotting them according to model size.


* In addition, it will also good to know what is the reference result, which is the result of original benchmark without any modifications. From existing results, it is hard to know how following instructions will impact models' content' correctness.


* The paper tiles has "using Knowledge Tasks" to evaluate model capabilities but I would only count MMLUPro as knowledge task and all others are reasoning related ones. In this case, the title of this paper should be further clarified.


* I do knot see too much value of Lite Benchmark version, which only consists of 150 samples. It introduces large variances when used for evaluating models. The authors should only consider have only one version for both simplicity and clarity, e.g. using the full benchmark or half size of the full version benchmark as the final benchmark.


References:

[1] Li et al. Instruction-following Evaluation through Verbalizer Manipulation. Findings of the Association for Computational Linguistics: NAACL 2024

**Questions:**

See the weakness.

---

> ### Author Response · Authors · 2024-11-23
> **Missing citation and Request to include more insights (Effect of Scale, Distractors within model families)**
>
> We would like to thank the reviewer for their detailed feedback and comments.
>
> Thank you for sharing the reference on the verbalizer paper - we had not come across this. Indeed our approach to study instruction following is based on verbalizers as a method. The reference does not include a benchmark and is based on label manipulation as a task.  We have updated the related to work to incorporate this reference.
>
> **Performance analysis with model family and distractors**
>
> We had previously included all models in different sizes and the intra-model family trends can be studied from the tables 4 in the paper but agree with the reviewer that trends may be hard to study with that view. Thank you for the feedback. We have now also included additional model family-specific figures in the appendix as per your suggestion.
>
> **On Applicable Instructions**: We report the performance on Full Benchmark for models from the Llama family and Qwen family of models in Figures in the appendix. We observe a consistent pattern of improvements in instruction following-ability with increase in model capacity for the Llama family. However, this is not the case for Qwen family of models. Specifically, for some instructions like *print_correct_answer, *print_correct_answer_label*, *sort_only_incorrect_answers the Qwen 1.5B model outperforms 3B model. Qwen 3B model is better than Qwen 7B and 14B variants for the *print_correct_answer_append_string* instruction. We consistently see 32B and 72B variants outperforming other models by a significant margin.
>
> **On inapplicable Instructions (distractors)**: We report details of how different model families (Llama, Qwen and Phi) are affected by distractors, at different scales. We find that the Llama family and Phi of models are extremely distracted by non-applicable instructions that require reversing and casing of text (eg: such instructions are inapplicable on numerical data), and report a drop of nearly 75-78\% while Qwen family of models (at all scales) is relatively robust to such distractors. On the other hand, distractor instructions that are based on numeric operations lead to a minor drop in performance in Llama and Qwen models but still affect Phi family of models significantly.

---

> ### Author Response · Authors · 2024-11-23
> **Comparison with reference result**
>
> We would like to clarify that the task of printing the correct answer label is an unperturbed assessment of the model performance. We demonstrate simply asking the model to print the text instead of the answer label results in a performance drop (See Figures 1) and are other instructions demonstrate that additional (simple) operations result in a further drop in performance.

---

> ### Author Response · Authors · 2024-11-23
> **Naming and Size of Lite Benchmark**
>
> **Naming**: Thank you for the feedback - we refer to knowledge tasks in the sense of verifiable grounding; perhaps grounding could be an alternative phrasing but that too has existing connotations in literature.
>
> **Lite Benchmark**: We would like to clarify that the Lite Benchmark has 150 samples from each dataset on which each instruction is applied i.e, we have 150 samples per instruction, for *each* dataset and for MMLUPro we have 25 samples per instruction. The total size of this benchmark still exceeds 12000 instances.

---

### Author Response · Authors · 2024-11-23
**Review Acknowledgement**

We would like to thank all reviewers for their insightful feedback and comments.  We are currently working on addressing all comments. To ensure a clear and timely response, we will post them incrementally as we complete the additional experiments and analyses requested.

---

### Author Response · Authors · 2024-11-27
**Addressed Comments**

Thank you all again for your feedback. We have attempted to address all your comments and would appreciate your thoughts / if you have any further comments.
We have updated our paper pdf based on your feedback and have highlighted the new content.

---

### Author Response · Authors · 2024-12-02
**Rebuttal Ending**

Dear Reviewers,

Thank you once again for your detailed reviews. We hope we've been able to address all your questions. Could we request you to please update your scores if that is the case. We'd be happy to answer any other questions you may have.

Thank you

---

### Meta-Review · Area_Chair_faFh · 2024-12-24

**Metareview:**

This paper considers the evaluation of instruction following abilities of LLMs using knowledge tasks. It modifies existing benchmarks by adding rules and evaluates the instruction following capabilities and correctness simultaneously.

Strength:

This paper develops benchmarks that evaluate both instruction following capabilities and content correctness jointly, which has not been studied before. It includes extensive experiments.

Weakness:

Several reviewers pointed out potential missing references and lack of discussion of the work in the proper context of existing works. In particular, it was also pointed by several reviewers that the paper does not have enough analysis and discussion of the results, making it hard to interpret the implications. The writing of the paper also needs significant work.

Overall, the paper still needs significant additional work and major revisions to make it publishable. Therefore, it cannot be published in the current state.

**Additional Comments On Reviewer Discussion:**

Reviewers does not follow up on the discussion. I have reviewed the author response to the reviewer comments. In particular, one points related to why evaluating instruction following and answer correctness has been successfully addressed. However, feedback in other aspects require the paper to go through a major revision and need to be reviewed again. Therefore, the paper cannot be accepted at this stage.

---

### Decision · Program_Chairs · 2025-01-22

Reject